# Layer-wise Sensitivity-aware Sparsity Allocation for Efficient LLM Inference

## Abstract

Large Language Model (LLM) inference presents substantial computational challenges when executed on commodity hardware, thereby necessitating the development of efficient acceleration techniques. While existing approaches predominantly focus on uniform compression strategies, they neglect the heterogeneous sensitivity patterns exhibited across different transformer layers. In this paper, we introduce Adaptive Sparsity Allocation Framework (ASAF), a novel approach that integrates rotation-based low-bit quantization with layer-wise adaptive sparsity allocation. The framework comprises two sequential phases with dynamic programming strategy. Phase 1: coarse-grained optimization that determines the optimal number of layer groups and narrows sparsity rate search intervals. Phase 2: fine-grained optimization that determines precise consecutive layer allocation and exact sparsity rates within each group. The joint optimization of layer grouping decisions and sparsity rate assignments creates a combinatorial explosion in the solution space, rendering brute-force approaches computationally prohibitive. To address this challenge, we employ a dynamic programming strategy that efficiently decomposes the exponential search space into manageable subproblems across both phases, achieving practical computational efficiency while guaranteeing global optimality. Extensive experiments conducted on the Llama-2 model family reveal that our proposed framework sustains benchmark accuracy degradation within 1%, concurrently achieving up to $3.63\times$ prefill acceleration and 12.63% memory reduction on NVIDIA RTX 3090 GPUs. This work advances beyond uniform compression strategies by recognizing and exploiting the distinct sensitivity characteristics of different transformer layers, thereby establishing a new paradigm for adaptive LLM compression on commodity hardware.

## 1 Introduction

LLMs underpin contemporary search, dialogue, and recommendation engines, driving a wide spectrum of user-facing AI services (Brown et al., 2020; Park et al., 2025). However, the computational demands of LLM inference impose substantial constraints on memory throughput and processing capacity of standard hardware accelerators (Frantar & Alistarh, 2023). A single inference pass alone can exhaust available GPU bandwidth, making each subsequent token generation a performance-critical bottleneck (Alizadeh et al., 2024). The sequential token-by-token generation inherent to autoregressive LLMs, where each prediction relies on the preceding context, establishes dependencies that inhibit parallel processing and amplify computational challenges (Korthikanti et al., 2023).

Recent research on efficient LLM inference has converged on two main directions: *i)* This direction focuses on low-bit quantization, which compresses activations and weights through orthogonal rotations and uniform low-bit encodings while preserving model accuracy (Dettmers et al., 2022; Ashkboos et al., 2024b; Xiao et al., 2023). These quantization approaches typically achieve substantial memory reduction and computational speedup, but often struggle with activation outliers that can significantly degrade model quality (Wei et al., 2022); *ii)* network sparsification through weight elimination, removing parameters to decrease computational workload (Han et al., 2015; Mishra et al., 2021; Dettmers et al., 2022). Although sparsification methods can attain high compression ratios with negligible performance degradation, existing strategies enforce identical sparsity levels across all network layers, disregarding the distinct sensitivity characteristics and compression responses exhibited by different architectural components, resulting in inefficient resource utilization (Kurtz et al., 2020). Nevertheless, these two approaches are fundamentally orthogonal, and current deployments usually have to choose one at the expense of the other (Sun et al., 2023).

Figure 1: (A) Traditional quantization methods and (B) our ASAF framework fusing quantization and sparsification, achieving 3.63× acceleration with $< 1\%$ precision loss.

In this paper, we propose ASAF that unifies rotation-based low-bit quantization and adaptive sparsity allocation through a two-phase optimization framework, as shown in Figure 1. Phase 1 performs coarse-grained optimization to determine optimal layer group numbers and sparsity intervals, while Phase 2 conducts fine-grained optimization for precise layer allocation and exact sparsity rates. The joint optimization creates combinatorial explosion in the solution space, which we address through dynamic programming that efficiently decomposes the exponential search space across both phases while guaranteeing global optimality. We implement our approach with customized low-bit kernels optimized for layer-grouped sparse operations (NVIDIA, 2023; Dao et al., 2022). ASAF addresses key limitations of existing approaches: rotation-based quantization mitigates activation outliers that plague low-bit methods, while adaptive layer-wise sparsity allocation ensures optimal compression capacity utilization unlike uniform pruning approaches. The main contributions are as follows:

- We identify an under-explored gap in current research where LLM inference optimization treats all layers uniformly during sparsification, ignoring the heterogeneous sensitivity patterns across transformer layers.

- We propose a novel framework that formulates layer-wise sparsity allocation as a constrained optimization problem with consecutive layer grouping constraints. We develop a two-phase approach that employs dynamic programming across both phases to efficiently decompose the exponential search space into manageable subproblems, achieving efficient optimization while guaranteeing global optimality.

- Our framework achieves up to 3.63× prefill speed-up and a 12.63% memory reduction on Llama-2-70B, with less than 1% degradation on standard language-understanding benchmarks, advancing the frontier of adaptive LLM compression on commodity GPUs.

## 2 MOTIVATION

In the deployment of LLMs, quantization and sparsification have evolved as independent acceleration techniques with limited integration. In our explorations, we conduct a series of attempts to investigate the fusion of both techniques. Figure 2(**Left**) demonstrates the inference acceleration achieved on the Llama-2-7B model in a language generation task, after applying 20% sparsity rate to various layers and layer combinations (weight matrices at 8-bit precision). We observe a significant improvement in model inference speed. Figure 2(**Right**) shows the model's performance across different quantization and sparsification configurations. As the bit-width of quantization decreases and the sparsity ratio increases, the perplexity rises substantially, indicating a clear trade-off between efficiency and performance. This observation gives rise to two key challenges:

**Precision-Compression Integration.** The introduction of quantization artifacts fundamentally reshapes weight significance patterns, rendering traditional sparsification strategies ineffective since they are originally developed for full-precision models (Dettmers et al., 2023; Nikolić et al., 2024). Simultaneously, sparsification operations modify activation characteristics, which subsequently disrupts optimal quantization parameter selection (Guo et al., 2024; Yao et al., 2022). This mutual interference necessitates a unified optimization framework that can systematically manage the complex interdependencies between these compression mechanisms (Guo et al., 2025).

**Solution Space Explosion.** The intersection of quantization and sparsification transforms the combinatorial optimization problem into a multidimensional challenge involving layer grouping decisions, sparsity rate assignments, and quantization interactions. This exponential growth in combinatorial possibilities renders brute-force search strategies computationally prohibitive (Choi & Cho, 2025; Dettmers et al., 2022; Frantar & Alistarh, 2023). Contemporary approaches suffer from evaluation overhead, demanding multiple fine-tuning cycles for performance assessment of each candidate configuration (Liu et al., 2025a; Ma et al., 2023; Mozafarinia et al., 2024). The critical need emerges for an efficient algorithm capable of identifying near-optimal layer grouping and sparsity allocation strategies within just a few epochs of low-precision fine-tuning (Zhang et al., 2022).

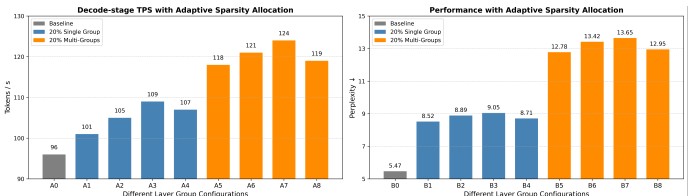

Figure 2: Performance analysis of Llama-2-7B across various layer group configurations. **Left**: Decode-stage speed improvement with uniform sparsity allocation across various configurations. **Right**: Quality degradation under different configurations. More details are in Appendix A.1.

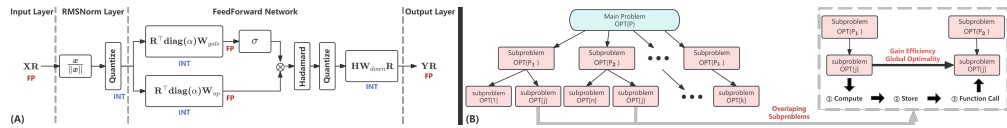

Figure 3: (A) Overview of rotation-based quantization framework for LLM transformer layers. (B) Illustration of dynamic programming approach showing problem decomposition into subproblems.

## 3 METHOD

### 3.1 BACKGROUND

**Quantization.** The low-bit quantization pipeline (as shown in Figure 3(A)) commences with the application of a randomized rotation matrix $R$ to the hidden state $X$ in floating-point precision (Ashkboos et al., 2023). This rotation operation disperses outlier activations, facilitating low-bit quantization processes (Ashkboos et al., 2024b). Following rotation, the state undergoes quantization to integer representation. The gate and up-projection weight matrices $W_{\text{gate}}$ and $W_{\text{up}}$ are pre-processed through multiplication with $R^T$ and scaling via the matrix $\text{diag}(\alpha)$, which encapsulates the absorbed RMSNorm scaling parameters. After processing through activation function $\sigma$, activations receive an online Hadamard transform (Tseng et al., 2024). The final stage employs a modified down-projection matrix $HW_{\text{down}}R$, where $H$ denotes the Hadamard transformation matrix and $R$ represents the rotation matrix, with quantization to integer precision followed by conversion to floating-point representation (Ashkboos et al., 2024a). More details are in Appendix A.2.

**Dynamic Programming.** Dynamic programming is an algorithmic paradigm that solves complex problems by breaking them down into simpler subproblems and storing the results to avoid redundant computations. The approach is applicable when a problem exhibits optimal substructure and overlapping subproblems, as shown in Figure 3(B). The optimal substructure property states that an optimal solution contains optimal solutions to its subproblems. Formally:

$$\text{OPT}(P) = f(\text{OPT}(P_1), \text{OPT}(P_2), \ldots, \text{OPT}(P_k)), \tag{1}$$

where $f$ combines the optimal solutions of subproblems to yield the optimal solution of the original problem. The overlapping subproblems property enables memoization to avoid redundant computations as shown in Figure 3(B). The general recurrence relation takes the form:

$$\text{OPT}[i] = \min_j \left\{ \text{OPT}[j] + \text{Cost}(j, i) \right\}, \tag{2}$$

where $\text{OPT}[i]$ represents the optimal solution for a subproblem of size $i$, and $\text{Cost}(j, i)$ denotes the cost of extending the solution from size $j$ to size $i$. More details can be found in Appendix A.3.

### 3.2 PROBLEM FORMULATION

In order to solve the challenges mentioned in the previous section, we formulate it as a constrained optimization problem. The primary objective is to find the optimal layer grouping and sparsity allocation strategy that minimizes total computational FLOPs while ensuring accuracy degradation remains within acceptable bounds. The mathematical formulation is defined as follows:

$$\{G^*, \{\mathcal{L}_i^*\}, \{s_i^*\}\} = \arg \min_{G, \{\mathcal{L}_i\}, \{s_i\}} \sum_{i=1}^{G} \sum_{l \in \mathcal{L}_i} \phi_l \times (1 - s_i), \tag{3}$$

where $G$ denotes the number of layer groups, $\mathcal{L}_i$ represents the set of consecutive layers in group

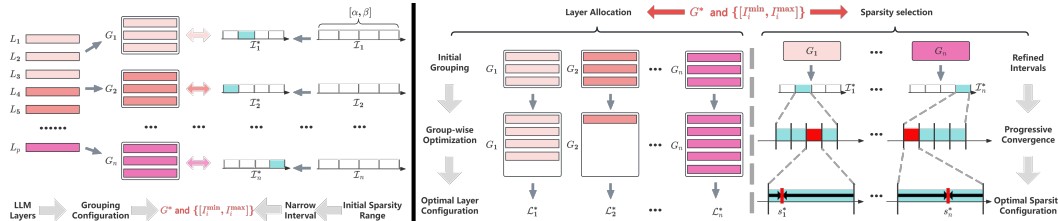

Figure 4: Overview of ASAF. **Left:** Coarse-grained optimization jointly determines optimal group number and narrows sparsity rate search intervals. **Right:** Fine-grained optimization jointly determines layer allocation within each group and sparsity rates. See Appendix A.4 for more details.

$i$, $s_i$ is the sparsity rate applied to group $i$, and $\phi_l$ represents the original computational cost (in FLOPs) of layer $l$. This optimization problem is subject to the following constraints:

$$\text{Accuracy degradation constraint: } \sum_{i=1}^{G} \xi(\mathcal{L}_i, s_i) \leq \delta_{\max}, \tag{4}$$

$$\text{Sparsity constraint: } \alpha \leq s_i \leq \beta, \quad \forall i \in \{1, 2, \ldots, G\}, \tag{5}$$

$$\text{Completeness constraint: } \bigcup_{i=1}^{G} \mathcal{L}_i = \{1, 2, \ldots, p\}, \quad \mathcal{L}_i \cap \mathcal{L}_j = \emptyset, \forall i \neq j, \tag{6}$$

$$\text{Continuity constraint: } \mathcal{L}_i = \{l_{\text{start}}^{(i)}, l_{\text{start}}^{(i)} + 1, \ldots, l_{\text{end}}^{(i)}\}, \tag{7}$$

where $\xi(\mathcal{L}_i, s_i)$ quantifies the layer sensitivity caused by applying sparsity rate $s_i$ to layer group $\mathcal{L}_i$, $\delta_{\max}$ represents the maximum allowable accuracy degradation threshold, $\alpha$ and $\beta$ define the lower and upper bounds of the feasible sparsity range, $l_{\text{start}}^{(i)}$ and $l_{\text{end}}^{(i)}$ denote the starting and ending layer indices of group $i$, $p$ denotes the total number of layers in the model, and the continuity constraint ensures consecutive sequences for efficient hardware batch processing, improved memory locality of adjacent layers, and significantly reduced combinatorial search space (Wang et al., 2023).

### 3.3 SOLUTION

To solve the formulated optimization problem, we propose the ASAF consisting of two sequential phases: coarse-grained and fine-grained optimization. Directly solving Equation 3 is computationally prohibitive due to the exponential search space created by joint grouping and sparsity decisions, so we decompose it into two manageable subproblems. The coarse-grained phase determines the optimal number of groups and narrows sparsity rate search intervals based on layer sensitivity, while the fine-grained phase determines the exact layer allocation within each group and the precise sparsity rates. In ASAF, we employ dynamic programming to efficiently explore the search space while maintaining the original objective of minimizing total computational FLOPs subject to accuracy degradation constraints. Different grouping configurations share common substructures, enabling memoization techniques to avoid redundant computations and achieve global optimality.

**Coarse-grained Optimization.** This phase jointly determines the optimal number of groups $G^*$ and refined sparsity intervals $\{[I_i^{\min}, I_i^{\max}]\}_{i=1}^{G^*}$ by evaluating all feasible group configurations and narrowing the search space $[\alpha, \beta]$ to focused sub-ranges for each group, as shown in Figure 4 (**Left**). This phase focuses on determining the optimal number of groups and narrowing the sparsity search space. The coarse-grained optimization solves:

$$\{G^*, \{I_i^*\}_{i=1}^{G^*}\} = \arg \min_{G, \{I_i\}_{i=1}^{G}} \left\{ \sum_{i=1}^{G} \min_{\mathcal{L}_i, s_i \in I_i} \sum_{l \in \mathcal{L}_i} \phi_l \times (1 - s_i) \right\}, \tag{8}$$

where $G$ represents the number of groups, $I_i$ denotes the sparsity interval for group $i$, $\mathcal{L}_i$ represents consecutive layers in group $i$, $s_i$ is the sparsity rate for group $i$, and $\phi_l$ is the original computational cost of layer $l$. The constraint ensures $\sum_{i=1}^{G} \xi(\mathcal{L}_i, s_i) \leq \delta_{\max}$, where $\xi(\mathcal{L}_i, s_i)$ quantifies the accuracy degradation caused by applying sparsity rate $s_i$ to layer group $\mathcal{L}_i$.

We define the dynamic programming state $\text{DP}_{\text{coarse}}[g][b]$ as the minimum total FLOP cost when using exactly $g$ groups with discretized accuracy degradation budget $b$:

$$\text{DP}_{\text{coarse}}[g][b] = \min_{\{I_i\}_{i=1}^g} \left\{ \sum_{i=1}^g \min_{\mathcal{L}_i, s_i \in I_i} \sum_{l \in \mathcal{L}_i} \phi_l \times (1 - s_i) \right\}, \tag{9}$$

where the state represents the optimal cost achievable with $g$ groups under accuracy degradation budget $b \times \Delta$ (with $\Delta$ being the discretization step), and the minimization considers all valid interval partitions and corresponding layer-sparsity assignments.

For each state, we consider all possible ways to add one more group by selecting an appropriate interval subset:

$$\text{DP}_{\text{coarse}}[g][b] = \min_{I \subseteq [\alpha, \beta]} \{ \text{OptimalCost}(I) + \text{DP}_{\text{coarse}}[g-1][b - \xi_{\text{cost}}(I)/\Delta] \}, \tag{10}$$

where $I$ represents the sparsity interval assigned to the $g$-th group, $\text{OptimalCost}(I)$ denotes the minimum FLOP cost achievable within interval $I$, and $\xi_{\text{cost}}(I)$ represents the accuracy degradation.

Algorithm 1 implements this optimization strategy by systematically exploring different group numbers while evaluating interval configurations through dynamic programming. The algorithm builds optimal solutions incrementally, with each state representing the minimum cost configuration for the given group count and accuracy degradation budget. The tabulation mechanism provides efficient access to FLOP costs and accuracy degradation values. The group number is explored within the range $G \in [1, p]$ where $p$ is the total number of layers, allowing for complete flexibility from single-group uniform allocation to maximum granularity with individual layer groups while maintaining computational tractability.

---

**Algorithm 1** Coarse-grained Optimization

---

**Require:** Layer count $p$, sparsity range $[\alpha, \beta]$, accuracy degradation threshold $\delta_{\max}$
**Ensure:** Optimal group number $G^*$ and intervals $\{[I_i^{\min}, I_i^{\max}]\}$
 1: Initialize $\text{DP}_{\text{coarse}}[0..p][0..\delta_{\max}/\Delta] \leftarrow \infty$, set boundary condition: $\text{DP}_{\text{coarse}}[0][0] \leftarrow 0$
 2: Generate interval candidates $\mathcal{I} = \{I : I \subseteq [\alpha, \beta]\}$
 3: **Main DP Loop:**
 4: **for** $g = 1$ **to** $p$ **do**
 5:    **for** $b = 0$ **to** $\delta_{\max}/\Delta$ **do**
 6:       **for** each interval $I \in \mathcal{I}$ **do**
 7:          $\text{cost} \leftarrow \text{OptimalCost}(I)$ using tabulation $H$
 8:          $\text{accuracy\_cost} \leftarrow \Xi_{\min}(I)/\Delta$ using tabulation $\Xi$
 9:          **if** $b - \text{accuracy\_cost} \geq 0$ **then**
10:             $\text{total\_cost} \leftarrow \text{cost} + \text{DP}_{\text{coarse}}[g-1][b - \text{accuracy\_cost}]$
11:             $\text{DP}_{\text{coarse}}[g][b] \leftarrow \min(\text{DP}_{\text{coarse}}[g][b], \text{total\_cost})$
12:          **end if**
13: **end for** (all nested loops)
14: **Solution Extraction:**
15: $G^* \leftarrow \arg\min_g \{\min_b \text{DP}_{\text{coarse}}[g][b]\}$
16: $\{I_i^{\min}, I_i^{\max}\} \leftarrow \text{BacktrackOptimalIntervals}(\text{DP}_{\text{coarse}}, G^*)$
17: **return** $G^*$, $\{[I_i^{\min}, I_i^{\max}]\}_{i=1}^{G^*}$

---

Upon completion, the coarse-grained optimization produces: (1) the optimal number of groups $G^*$ that minimizes computational overhead while satisfying accuracy constraints, and (2) refined sparsity intervals $\{[I_i^{\min}, I_i^{\max}]\}_{i=1}^{G^*}$ that narrow the search space from the original range $[\alpha, \beta]$.

**Rationale for Layer Grouping.** Consecutive layer-grouping in ASAF is motivated not solely by computational concerns but by architectural regularities in Transformer models. Adjacent blocks often share the same hidden dimension, attention/MLP structure and residual pathway pattern, which leads to slowly varying sensitivity to sparsity rather than abrupt layer-by-layer differences. Numerous prior works on pruning and structural compression (Men et al., 2025; Ma et al., 2023; Lu et al., 2024) observe that mid-depth layers behave as interchangeable units while the first few and final layers are more fragile. By grouping consecutive layers, ASAF captures this smooth depth-wise sensitivity profile, reducing redundancy in the search space while retaining flexibility: the subsequent fine-grained stage can allocate distinct sparsity within each group if needed.

**Fine-grained Optimization.** This phase receives the optimal group number $G^*$ and refined intervals $\{[I_i^{\min}, I_i^{\max}]\}_{i=1}^{G^*}$ from the coarse-grained phase, then jointly determines the exact consecutive layer allocation $\{\mathcal{L}_i^*\}_{i=1}^{G^*}$ and precise sparsity rates $\{s_i^*\}_{i=1}^{G^*}$, as shown in Figure 4 (**Right**).

Given the optimal group number $G^*$ and refined intervals $\{[I_i^{\min}, I_i^{\max}]\}_{i=1}^{G^*}$ from the coarse-grained phase, this phase determines the exact layer allocation and precise sparsity rates:

$$\{\{\mathcal{L}_i^*\}_{i=1}^{G^*}, \{s_i^*\}_{i=1}^{G^*}\} = \arg \min_{\{\mathcal{L}_i\}_{i=1}^{G^*}, \{s_i\}_{i=1}^{G^*}} \left\{ \sum_{i=1}^{G^*} \sum_{l \in \mathcal{L}_i} \phi_l \times (1 - s_i) \right\}, \tag{11}$$

where $\{\mathcal{L}_i\}_{i=1}^{G^*}$ represents the layer allocation with each $\mathcal{L}_i$ containing consecutive layers, and $\{s_i\}_{i=1}^{G^*}$ denotes the sparsity rates. The constraints ensure $\sum_{i=1}^{G^*} \xi(\mathcal{L}_i, s_i) \leq \delta_{\max}$, $s_i \in [I_i^{\min}, I_i^{\max}]$, $\mathcal{L}_i$ are consecutive, $\bigcup_{i=1}^{G^*} \mathcal{L}_i = \{1, 2, \ldots, p\}$, and $\mathcal{L}_i \cap \mathcal{L}_j = \emptyset$ for $i \neq j$.

We define the dynamic programming state $\mathrm{DP}_{\text{fine}}[i][g][b]$ as the minimum total FLOP cost for optimally partitioning layers $[i..p]$ into exactly $g$ consecutive groups with remaining accuracy degradation budget $b$:

$$\mathrm{DP}_{\text{fine}}[i][g][b] = \min_{\{\mathcal{L}_k\}_{k=1}^g, \{s_k\}_{k=1}^g} \left\{ \sum_{k=1}^{g} \sum_{l \in \mathcal{L}_k} \phi_l \times (1 - s_k) \right\}, \tag{12}$$

where the state covers layers from position $i$ to $p$, requires exactly $g$ groups, has remaining budget $b$, and ensures $\bigcup_{k=1}^{g} \mathcal{L}_k = \{i, i+1, \ldots, p\}$ with each $\mathcal{L}_k$ consecutive, $s_k \in [I_k^{\min}, I_k^{\max}]$, and $\sum_{k=1}^{g} \xi(\mathcal{L}_k, s_k) \leq b$.

For each state, we jointly enumerate all possible first-group formations and sparsity assignments within the corresponding refined interval. The state transition equation is:

$$\mathrm{DP}_{\text{fine}}[i][g][b] = \min_{\substack{j \in [i, p-g+1] \\ s \in [I_g^{\min}, I_g^{\max}]}} \left\{ \mathrm{GroupCost}(i, j, s) + \mathrm{RemainingCost}(j+1, g-1, b') \right\}, \tag{13}$$

where the total cost consists of two components. (1) The current group cost $\mathrm{GroupCost}(i, j, s) = H[i][j - i + 1][s]$ represents the FLOP cost for assigning layers $i$ to $j$ with sparsity rate $s$, retrieved from pre-computed tabulation. (2) The remaining subproblem cost $\mathrm{RemainingCost}(j+1, g-1, b') = \mathrm{DP}_{\text{fine}}[j+1][g-1][b']$ represents the optimal cost for the remaining layers $[j+1, p]$ using $g-1$ groups with updated budget $b' = b - \lceil \Xi[i][j-i+1][s]/\Delta \rceil$, where $\Xi[i][j-i+1][s]$ gives the accuracy degradation cost from tabulation.

---

**Algorithm 2** Fine-grained Optimization

---

**Require:** $G^*$ groups, intervals $\{[I_i^{\min}, I_i^{\max}]\}$, tabulation tables $H$, $\Xi$
**Ensure:** Optimal allocation $\{\mathcal{L}_i^*\}$ and sparsity rates $\{s_i^*\}$
 1: Initialize $\mathrm{DP}_{\text{fine}}[1..p+1][0..G^*][0..\delta_{\max}/\Delta] \leftarrow \infty$, $\mathrm{choice}[1..p+1][0..G^*][0..\delta_{\max}/\Delta] \leftarrow \emptyset$
 2: Set boundary condition: $\mathrm{DP}_{\text{fine}}[p+1][0][b] \leftarrow 0$ for all $b \geq 0$
 3: **Backward DP Construction:**
 4: **for** $i = p$ **down to** $1$ **do**
 5:    **for** $g = 1$ **to** $\min(G^*, p - i + 1)$ **do**
 6:       **for** $b = 0$ **to** $\delta_{\max}/\Delta$ **do**
 7:          **for** $j = i$ **to** $p - g + 1$, $s \in \mathrm{Discretize}([I_{G^*-g+1}^{\min}, I_{G^*-g+1}^{\max}])$ **do**
 8:             group_cost, accuracy_cost $\leftarrow H[i][j - i + 1][s]$, $\Xi[i][j - i + 1][s]/\Delta$
 9:             **if** $b - \mathrm{accuracy\_cost} \geq 0$ **then**
10:                total_cost $\leftarrow$ group_cost $+ \mathrm{DP}_{\text{fine}}[j+1][g-1][b - \mathrm{accuracy\_cost}]$
11:                **if** total_cost $< \mathrm{DP}_{\text{fine}}[i][g][b]$ **then**
12:                   $\mathrm{DP}_{\text{fine}}[i][g][b] \leftarrow \mathrm{total\_cost}$, $\mathrm{choice}[i][g][b] \leftarrow (j, s)$
13:             **end if** (nested conditions)
14: **end for** (all nested loops)
15: **Solution Reconstruction:**
16: $\{\mathcal{L}_i^*, s_i^*\} \leftarrow \mathrm{BacktrackSolution}(\mathrm{choice}, 1, G^*, \delta_{\max}/\Delta)$
17: **return** $\{\mathcal{L}_i^*\}_{i=1}^{G^*}$, $\{s_i^*\}_{i=1}^{G^*}$

---

Algorithm 2 implements the fine-grained optimization through backward dynamic programming construction. The algorithm determines precise layer boundaries and sparsity assignments by working from final layers toward initial layers, ensuring each decision considers all downstream implications while respecting refined interval constraints from the coarse-grained phase.

Upon completion, the two-phase ASAF framework delivers a complete solution: optimal group number $G^*$, precise consecutive layer allocation $\{\mathcal{L}_i^*\}_{i=1}^{G^*}$, and adaptive sparsity rates $\{s_i^*\}_{i=1}^{G^*}$ that jointly minimize computational FLOPs while maintaining accuracy constraints.

**Tabulation.** The tabulation mechanism provides efficient access to FLOPs and accuracy degradation through pre-computed tables: $H[i][\text{len}][s]$ stores FLOP costs for consecutive layers starting from position $i$ with length len under sparsity rate $s$, while $\Xi[i][\text{len}][s]$ records corresponding accuracy degradation costs. These tables enable $O(1)$ lookup during dynamic programming transitions, transforming expensive evaluations into efficient queries. Construction details are in Appendix A.7.

## 4 EXPERIMENT

### 4.1 IMPLEMENTATION DETAILS

**Software and Hardware Setup.** Our ASAF framework is built upon QuaRot (Ashkboos et al., 2024b) using PyTorch (Paszke et al., 2019) with CUDA-12.1 (NVIDIA, 2023), evaluated on Llama-2 family models (Touvron et al., 2023). We implement custom CUDA kernels for dynamic programming tabulation and layer-wise accuracy degradation measurement, with all evaluations conducted on NVIDIA RTX 3090 GPUs. More details can be found in Appendix A.8.

**Framework Hyperparameters.** We explore sparsity allocations where $\alpha = 1\%$ and $\beta = 15\%$ define the feasible sparsity range. The maximum allowable accuracy degradation is set to $\delta_{\max} = 1\%$ with tabulation resolution $\Delta = 0.5\%$. More details can be found in Appendix A.8.

**Experimental Configuration.** We employ 4-bit GPTQ quantization (Frantar et al., 2022) for weights with group size 128, symmetric per-token quantization for activations (Xiao et al., 2023), and asymmetric quantization for KV caches (Dettmers et al., 2022). All configurations utilize clipping ratios of 0.9-0.95 and maintain numerical stability through FP32 accumulation during tabulation construction. More details can be found in Appendix A.8.

**Precomputation Overhead.** To clarify the computational cost of ASAF, we measure the time required to construct the tabulation tables $H$ and $\Xi$ and to run the dynamic-programming search. The tabulation process scales linearly with the number of layers and the sparsity grid and does not require any additional training. On Llama-2-70B, generating all table entries on a single RTX 3090 takes 2.6 GPU-hours, and the subsequent DP search finishes in under three minutes, while smaller models incur proportionally lower cost. This preprocessing is performed only once for each model, after which the resulting sparsity allocation can be reused across deployments without further computation. Given the substantial inference-time savings enabled by ASAF, this one-time overhead is negligible in practical applications. Deeper architectures would incur a roughly proportional increase in this one-time preprocessing cost, which remains negligible compared to the long-term inference-time savings. The resulting sparsity configuration can then be reused across deployments without further computation.

### 4.2 ACCURACY ANALYSIS

**Language Generation Tasks.** We evaluate our ASAF framework on the WikiText-2 language-generation benchmark. Table 1 reports the perplexity after quantizing Llama-2 weights to 4 bits with GPTQ and applying our adaptive sparsity allocation across layer groups. Our framework demonstrates competitive performance compared to recent advanced quantization methods, achieving perplexity degradation less than 1% compared to QuaRot while providing additional computational benefits through optimized sparsity allocation. The layer-group-based pruning approach requires no additional outlier storage or asymmetric quantization schemes. When using group-size-128 quantization, ASAF maintains comparable performance with perplexity increases within 1% of QuaRot-128G while enabling more efficient inference through adaptive sparsity patterns.

**Zero-Shot Tasks.** We assess ASAF across six established zero-shot benchmarks: PIQA (Bisk et al., 2020), WinoGrande (Sakaguchi et al., 2021), HellaSwag (Zellers et al., 2019), LAMBADA (Radford et al., 2019), and ARC-Easy and ARC-Challenge (Clark et al., 2018). Experiments utilize the

Table 1: WikiText-2 perplexity comparison for Llama-2 models (2048 sequence length) using 4-bit quantization with adaptive sparsity allocation. SmoothQuant and OmniQuant results are from (Shao et al., 2023), and 128G indicates group-wise quantization with 128 group size. More details in Appendix A.9.

| Method | Weight Quantization | #Outlier Features | Llama-2 7B | 13B | 30B | 70B |
|---|---|---|---|---|---|---|
| Baseline | - | - | 5.47 | 4.88 | 4.09 | 3.32 |
| SmoothQuant (Xiao et al., 2023) | RTN | 0 | 83.12 | 35.88 | - | - |
| OmniQuant (Shao et al., 2023) | RTN | 0 | 14.26 | 12.30 | - | - |
| QUIK-4B (Ashkboos et al., 2023) | GPTQ | 256 | 8.87 | 7.78 | 7.28 | 6.91 |
| QuaRot (Ashkboos et al., 2024b) | GPTQ | 0 | **6.10** | **5.40** | **4.41** | **3.79** |
| ASAF (Ours) | GPTQ | 0 | 6.14 | 5.44 | 4.44 | 3.82 |
| Atom-128G (Zhao et al., 2023) | | 128 | 6.03 | **5.26** | - | - |
| QuaRot-128G (Ashkboos et al., 2024b) | GPTQ-128G | 0 | **5.93** | **5.26** | **4.25** | **3.61** |
| ASAF-128G (Ours) | | 0 | 5.98 | 5.30 | 4.28 | 3.64 |

Table 2: Zero-shot accuracy of Llama models with our ASAF framework on PIQA (PQ), WinoGrande (WG), HellaSwag (HS), Arc-Easy (A-e), Arc-Challenge (A-c), and LAMBADA (LA).

| Model | Method | PQ ↑ | WG ↑ | HS ↑ | A-e ↑ | A-c ↑ | LA ↑ | Avg. ↑ |
|---|---|---|---|---|---|---|---|---|
| Llama2-7B | FP16 | 79.11 | 69.06 | 75.99 | 74.58 | 46.25 | 73.90 | 69.82 |
| | QuaRot | 76.77 | 63.77 | 72.16 | 69.87 | 40.87 | 70.39 | 65.64 |
| | ASAF (Ours) | 76.00 | 63.23 | 71.47 | 69.17 | 40.52 | 69.69 | 65.01 |
| Llama2-13B | FP16 | 80.47 | 72.22 | 79.39 | 77.48 | 49.23 | 76.75 | 72.59 |
| | QuaRot | 78.89 | 70.24 | 76.37 | 72.98 | 46.59 | 73.67 | 69.79 |
| | ASAF (Ours) | 78.26 | 69.68 | 75.76 | 72.25 | 46.19 | 72.97 | 69.19 |
| Llama2-30B | FP16 | 81.13 | 73.94 | 80.72 | 78.52 | 51.65 | 77.59 | 73.93 |
| | QuaRot | 79.94 | 72.03 | 77.99 | 75.20 | 49.46 | 75.18 | 71.63 |
| | ASAF (Ours) | 79.14 | 71.42 | 77.37 | 74.60 | 48.99 | 74.58 | 71.02 |
| Llama2-70B | FP16 | 82.70 | 77.98 | 83.84 | 80.98 | 57.34 | 79.58 | 77.07 |
| | QuaRot | 82.43 | 76.24 | 81.82 | 80.43 | 56.23 | 78.73 | 75.98 |
| | ASAF (Ours) | 81.77 | 75.48 | 81.12 | 79.71 | 55.81 | 77.98 | 75.31 |

Table 3: GPU memory consumption (MB) of QuaRot and ASAF across Llama-2 and Llama-3 models with varying sequence lengths. Comp. Ratio denotes compression ratio.

| Model | Method | Sequence Length 512 | 2048 | 4096 |
|---|---|---|---|---|
| Llama-2-7B | QuaRot | 1,019 | 3,255 | 6,518 |
| | ASAF | 948 | 3,013 | 6,009 |
| | Comp. Ratio | 6.97% | 7.43% | 7.81% |
| Llama-2-13B | QuaRot | 1,849 | 5,753 | 11,486 |
| | ASAF | 1,688 | 5,276 | 10,502 |
| | Comp. Ratio | 8.71% | 8.29% | 8.57% |
| Llama-2-30B | QuaRot | 3,573 | 11,408 | 22,814 |
| | ASAF | 3,127 | 10,110 | 20,148 |
| | Comp. Ratio | 12.48% | 11.38% | 11.69% |
| Llama-2-70B | QuaRot | 6,521 | 20,536 | 41,079 |
| | ASAF | 5,653 | 17,943 | 35,833 |
| | Comp. Ratio | 13.31% | 12.63% | 12.77% |
| Llama-3-8B | QuaRot | 1,226 | 3,722 | 7,456 |
| | ASAF | 1,127 | 3,432 | 6,869 |
| | Comp. Ratio | 8.08% | 7.79% | 7.87% |
| Llama-3-70B | QuaRot | 6,811 | 21,295 | 42,586 |
| | ASAF | 5,855 | 18,136 | 36,241 |
| | Comp. Ratio | 14.04% | 14.83% | 14.90% |

Table 4: WikiText-2 perplexity (PPL) and zero-shot accuracy of our ASAF framework for Llama-2 models applying 4- and 8-bits with RTN weights and activation quantization. More details can be found in Appendix A.9.

| Model | Method | Prec. | PPL ↓ | PQ ↑ | WG ↑ | HS ↑ | A-e ↑ | A-c ↑ | LA ↑ | Avg. ↑ |
|---|---|---|---|---|---|---|---|---|---|---|
| 7B | Baseline | FP16 | 5.47 | 79.11 | 69.06 | 75.99 | 74.58 | 46.25 | 73.90 | 69.82 |
| | QuaRot-RTN | INT4 | 8.37 | 72.09 | 60.69 | 65.40 | 58.88 | 35.24 | 57.27 | 58.26 |
| | | INT8 | 5.50 | 78.94 | 68.67 | 75.80 | 74.79 | 45.39 | 74.33 | 69.65 |
| | ASAF-RTN (Ours) | INT4 | 8.44 | 71.48 | 60.23 | 64.81 | 58.38 | 34.98 | 56.75 | 57.77 |
| | | INT8 | 5.54 | 78.31 | 68.09 | 75.12 | 74.19 | 45.00 | 73.66 | 69.06 |
| 13B | Baseline | FP16 | 4.88 | 80.47 | 72.22 | 79.39 | 77.48 | 49.23 | 76.75 | 72.59 |
| | QuaRot-RTN | INT4 | 6.09 | 77.37 | 67.32 | 73.11 | 70.83 | 43.69 | 70.66 | 67.16 |
| | | INT8 | 4.90 | 80.52 | 71.59 | 79.38 | 77.31 | 49.15 | 76.79 | 72.46 |
| | ASAF-RTN (Ours) | INT4 | 6.14 | 76.67 | 66.71 | 72.45 | 70.19 | 43.30 | 70.02 | 66.56 |
| | | INT8 | 4.94 | 79.84 | 70.91 | 78.67 | 76.65 | 48.68 | 76.10 | 71.81 |
| 30B | Baseline | FP16 | 4.42 | 81.13 | 73.94 | 80.72 | 78.52 | 51.65 | 77.59 | 73.93 |
| | QuaRot-RTN | INT4 | 5.51 | 78.36 | 69.65 | 75.05 | 72.84 | 46.08 | 72.56 | 69.09 |
| | | INT8 | 4.43 | 81.18 | 73.35 | 80.65 | 78.36 | 51.58 | 77.63 | 73.82 |
| | ASAF-RTN (Ours) | INT4 | 5.56 | 77.69 | 68.99 | 74.37 | 72.22 | 45.64 | 71.91 | 68.47 |
| | | INT8 | 4.47 | 80.45 | 72.69 | 79.92 | 77.65 | 51.12 | 76.93 | 73.13 |
| 70B | Baseline | FP16 | 3.32 | 82.70 | 77.98 | 83.84 | 80.98 | 57.34 | 79.58 | 77.07 |
| | QuaRot-RTN | INT4 | 4.14 | 80.69 | 75.14 | 79.63 | 77.57 | 51.71 | 77.02 | 73.63 |
| | | INT8 | 3.33 | 82.97 | 77.98 | 83.67 | 80.77 | 58.11 | 79.53 | 77.17 |
| | ASAF-RTN (Ours) | INT4 | 4.18 | 79.92 | 74.46 | 78.83 | 76.83 | 51.24 | 76.25 | 72.92 |
| | | INT8 | 3.36 | 82.14 | 77.24 | 82.92 | 79.96 | 57.56 | 78.81 | 76.44 |

LM Evaluation Harness (Gao et al., 2021; 2024) with default configurations. Table 2 shows that ASAF maintains strong performance across all Llama-2 model sizes, with performance degradation consistently below 1% compared to QuaRot. The ASAF preserves model capabilities while enabling computational efficiency gains through optimized layer-group pruning patterns.

## 4.3 Performance Analysis

**Prefill Stage Performance Increases.** Figure 5 demonstrates the prefill-stage acceleration of ASAF across various batch configurations (1, 4, 16, and 32) with 2048-token sequences on Llama-2 models. Our adaptive sparsity allocation approach consistently outperforms the QuaRot baseline across all configurations. The performance gains become more pronounced with larger batch sizes, as the computational workload increasingly overshadows memory bandwidth limitations. For the largest 70B model, our method reaches peak acceleration of 3.63×. The results reveal a trend where both increasing model complexity and batch size magnify the effectiveness of our strategy, demonstrating the scalable nature of the ASAF framework's adaptive sparsity allocation mechanism.

**Memory and Computational Efficiency.** The algorithmic efficiency of ASAF directly translates into substantial memory savings across both model families and sequence lengths. Table 3 expands our evaluation beyond the Llama-2 series to include the more recent Llama-3 models and reports results under three context lengths: 512, 2048, and 4096 tokens. This broader evaluation shows that ASAF maintains stable compression ratios across architectures and sequence lengths, achieving 7 to 15% memory reduction depending on model scale, and confirming that its sensitivity-aware sparsity allocation generalizes beyond a single model generation. The ability to preserve compression efficacy on Llama-3 is particularly noteworthy given architectural and training differences that often affect structured sparsification. Furthermore, the multi-length analysis addresses prefill-related concerns: ASAF consistently reduces memory usage even at short (512) and long (4096) contexts, providing favorable conditions for high-throughput prefill optimization. Additional computational-efficiency results, including kernel-level analysis, are provided in Appendix A.10.

Figure 5: Prefill-stage acceleration comparison of ASAF framework versus QuaRot on Llama-2 models using NVIDIA RTX 3090 GPUs with 2048-token sequences across different batch sizes.

Table 5: End-to-end throughput and latency on RTX 3090 for Llama-2-7B and Llama-2-13B. Higher tokens/s (↑) and lower latency (↓) indicate better performance.

| Model | Prompt length | Gen. length | Batch size | Method | Tokens/s (↑) | Latency (ms) (↓) |
|---|---|---|---|---|---|---|
| Llama-2-7B | 512 | 128 | 1 | QuaRot | 909 | 711 |
| | | | | ASAF | 945 | 677 |
| | 2048 | 512 | 1 | QuaRot | 703 | 3657 |
| | | | | ASAF | 805 | 3180 |
| Llama-2-13B | 2048 | 512 | 8 | QuaRot | 2611 | 7877 |
| | | | | ASAF | 3257 | 6302 |
| | 4096 | 512 | 8 | QuaRot | 2129 | 17554 |
| | | | | ASAF | 2716 | 13503 |

Table 6: Comparison of downstream task performance on LLaMA-7B across WikiText-2 and seven zero-shot benchmarks: PIQA (PQ) (Bisk et al., 2020), WinoGrande (WG) (Sakaguchi et al., 2021), HellaSwag (HS) (Zellers et al., 2019), Arc-Easy (A-e) and Arc-Challenge (A-c) (Clark et al., 2018), BoolQ (Clark et al., 2019), and MMLU (Hendrycks et al., 2021). Lower WikiText perplexity (↓) is better; higher accuracy scores (↑) are better.

| Method | WikiText ↓ | PQ ↑ | WG ↑ | HS ↑ | A-e ↑ | A-c ↑ | BoolQ ↑ | MMLU ↑ | Avg. ↑ |
|---|---|---|---|---|---|---|---|---|---|
| SpinQuant (Liu et al., 2025b) | 5.94 | 76.14 | 69.58 | 72.16 | 69.30 | 40.37 | 63.77 | 30.00 | 60.19 |
| OWL (Yin et al., 2025) | 24.55 | 62.48 | 58.72 | 44.79 | 45.03 | 26.19 | 58.48 | 26.84 | 46.08 |
| DSA (Li et al., 2024) | 12.50 | 68.13 | 65.97 | 61.84 | 65.26 | 39.42 | 66.84 | 31.05 | 56.93 |
| SLIM (Mozaffari et al., 2025) | 9.80 | 75.04 | 67.83 | 67.35 | 67.14 | 41.38 | 70.23 | 29.41 | 59.77 |
| ASAF (Ours) | 5.98 | 76.00 | 63.23 | 71.47 | 69.17 | 40.52 | 74.23 | 31.87 | 60.93 |

**End-to-end Latency and Throughput.** We measure full-sequence inference performance including both the prompt processing and autoregressive decoding phases to complement the prefill-focused results. We evaluate Llama-2-7B and Llama-2-13B models on an RTX 3090 GPU under several realistic configurations combining prompt lengths of 512, 2048, and 4096 tokens, generation lengths of 128 and 512 tokens, and batch sizes of 1 and 8. Table 5 reports the end-to-end tokens-per-second and wall-clock latency for the QuaRot baseline and our ASAF-optimized models. Across all tested settings, ASAF improves end-to-end throughput while respecting the same ≤1% accuracy budget used throughout the paper. Because ASAF targets the large matrix multiplications that dominate the prefill stage while leaving the lightweight decoding computations unchanged, its impact naturally varies with the relative cost of these phases: the gains are most pronounced for long prompts and large batches, where prefill dominates the overall inference time. For short prompts, where decoding contributes a larger fraction of runtime, the improvements are moderate but consistently positive, confirming that ASAF introduces no decoding-phase overhead.

**Layer Sensitivity and Sparsity Pattern Analysis.** We examine the sparsity distribution learned by ASAF to understand how different parts of the model contribute to sensitivity. Across all Llama-2 models, we observe a consistent pattern: early blocks close to the embedding layer and the final decoder blocks tend to be more sensitive and therefore receive lower sparsity, while mid-depth blocks tolerate higher sparsity, especially the MLP sublayers. This observation aligns with widely reported behaviors in Transformer architectures: early layers shape core token representations, final layers refine task-specific semantics, and mid-layer MLPs contain the largest amount of structural redundancy. ASAF captures these properties automatically, without hand-crafted heuristics.

**Comparison with Recent Compression Methods.** Table 6 presents a comparative analysis of ASAF against recent compression approaches on LLaMA-7B. ASAF achieves competitive WikiText-2 perplexity (5.98 vs. SpinQuant's 5.94), with negligible degradation of 0.04 points. More importantly, ASAF attains the highest average accuracy (60.93%) across seven zero-shot benchmarks, outperforming SpinQuant (60.19%), SLIM (59.77%), and other baselines. In contrast, pure sparsification methods suffer from substantial quality degradation, with OWL and DSA exhibiting perplexities of 24.55 and 12.50 respectively, alongside significantly lower task accuracies (46.08% and 56.93%). These results demonstrate that ASAF's layer-wise sensitivity-aware sparsity allocation, when integrated with rotation-based quantization, achieves superior accuracy-efficiency trade-offs compared to uniform compression strategies or standalone sparsification techniques.

**Prefill Stage Performance on NVIDIA 4090 GPU.** The Figure in Appendix A.11 (due to page limit) reports prefill acceleration experiments across different models and methods. ASAF achieves up to 3.89× acceleration on Llama-2-70B, validating hardware scalability.

**Zero-Shot Tasks on Llama-3 Family.** The Table in Appendix A.12 (due to page limit) reports zero-shot evaluation across different models and methods. ASAF demonstrates generalizability and consistently maintains <1% performance degradation versus QuaRot.

### 4.4 ABLATION STUDIES

**RTN Quantization Strategy.** To evaluate the robustness of our adaptive sparsity allocation approach, we compare ASAF's performance under RTN quantization, where GPTQ serves as the default weight quantization strategy. Table 4 demonstrates that at 8-bit precision, RTN maintains accuracy nearly identical to full precision for both approaches. At 4-bit quantization, while both methods experience some quality degradation, our ASAF framework consistently maintains performance within 1% of QuaRot results across all model sizes. In both INT4 and INT8 configurations, these findings confirm that layer-group-based adaptive sparsity allocation can be effectively combined with RTN quantization without introducing significant accuracy loss, validating the generalizability of our optimization framework.

**Group-Wise Quantization.** The Table in Appendix A.13 (due to page limit) reports WikiText-2 perplexity for our ASAF framework when weights and activations are quantized group-wise with group sizes of 256, 128, and 64. As expected, smaller groups yield better accuracy because per-group scale factors more precisely capture local statistics, though they incur additional scale storage and slightly more complex kernels. Across every group size, our adaptive sparsity allocation framework tracks QuaRot's dense counterparts to within 1%, demonstrating that layer-group-based sparsity optimization can be achieved without meaningful quality loss. The consistent performance across different group sizes validates the robustness of our ASAF approach under various quantization granularities.

## 5 RELATED WORK

Recent advances in LLM compression have significantly improved inference efficiency while preserving model capabilities. Structured pruning techniques eliminate entire components like attention heads or layers, providing coarse-grained compression with predictable memory reduction but limited optimization flexibility (Dutta et al., 2024; Muralidharan et al., 2024). Weight quantization methods reduce numerical precision while maintaining computational accuracy, with notable frameworks including AWQ (Lin et al., 2024), SpinQuant (Liu et al., 2025b), and OmniQuant (Shao et al., 2023). Knowledge distillation approaches like MiniLLM (Gu et al., 2023) and GKD (Agarwal et al., 2024) enable effective knowledge transfer from larger teacher models to compact student networks.

Unstructured pruning offers flexibility by removing weights, enabling fine-grained sparsity patterns that achieve better accuracy-efficiency trade-offs at high compression ratios. Recent techniques include SparseGPT (Frantar & Alistarh, 2023), applying layer-wise reconstruction using second-order approximations, Wanda (Sun et al., 2023), using activation-aware magnitude selection, and advanced methods like OWL (Yin et al., 2025), ALPS (Meng et al., 2024), and DSnoT (Zhang et al., 2023). Model quantization addresses the critical outlier problem where activations dominate quantization ranges. Rotation-based approaches like QuaRot (Ashkboos et al., 2024b) and QuIP (Chee et al., 2023) employ orthogonal transformations to redistribute outlier energy, while SmoothQuant (Xiao et al., 2023) migrates difficulty from activations to weights. System optimizations include FlashAttention (Dao et al., 2022), PagedAttention (Kwon et al., 2023), and inference frameworks like SGLang (Zheng et al., 2024) and vLLM (Kwon et al., 2023). Recent work (Yuan et al., 2025) demonstrates training-time attention sparsity through hardware-aligned kernels. However, existing approaches treat quantization and pruning independently without considering layer sensitivity.

## 6 CONCLUSION

We present the ASAF that systematically allocates sparsity across layer groups to minimize computational FLOPs while maintaining model accuracy. Our two-phase strategy employs dynamic programming with tabulation to achieve efficient optimization with manageable computational complexity, making large-scale optimization tractable for practical deployment. Experimental results demonstrate that ASAF maintains accuracy degradation within 1% while achieving up to 3.63× prefill acceleration and 12.63% memory reduction on Llama-2 models compared to QuaRot.

ETHICS STATEMENT

This work adheres to the ICLR Code of Ethics. Our research focuses on developing efficient inference methods for large language models. We identify the following ethical considerations:

**Privacy.** No personally identifiable information is collected or processed.

**Environmental Impact.** While our method reduces computational costs and energy consumption compared to baseline approaches, LLM inference still requires significant computational resources. We report detailed computational requirements in Appendix A.8.

**Potential Harms.** Our compression technique could potentially be applied to harmful applications. We emphasize the importance of responsible deployment and adherence to AI safety guidelines.

REPRODUCIBILITY STATEMENT

To facilitate reproduction of our results:

**Code.** We will release our complete implementation, including training scripts, evaluation code, and CUDA kernels, upon paper acceptance to facilitate reproduction of our results.

**Experimental Details.** Hyperparameters and experimental setup are fully specified in Appendix A.8. Hardware specifications are provided in Appendix A.8.

**Data.** We use publicly available datasets and model checkpoints. Datasets: WikiText-2 for language modeling perplexity evaluation; PIQA, WinoGrande, HellaSwag, LAMBADA, ARC-Easy and ARC-Challenge for zero-shot reasoning tasks; BoolQ and MMLU for downstream task evaluation. Models: Llama-2-7B, Llama-2-13B, Llama-2-30B, and Llama-2-70B; Llama-3-8B and Llama-3-70B. All datasets and pre-trained model checkpoints are publicly accessible.

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

## A  APPENDIX

All appendices are provided in the supplementary text.

