# Layer-wise Sensitivity-aware Sparsity Allocation for Efficient LLM Inference

## The Use of Large Language Models (LLMs)

We use Claude 4 Sonnet and ChatGPT 5 for grammar checking, spelling correction, and translation assistance in both the main text and appendix of this paper.

## Related Work

Recent advances in LLM compression have significantly improved inference efficiency while preserving model capabilities. Structured pruning techniques eliminate entire components like attention heads or layers (Dutta et al., 2024; Muralidharan et al., 2024), with approaches like TVAPrune leveraging variational information bottleneck principles to compress model representations (Dutta et al., 2024). Weight quantization methods such as GPTQ (Frantar et al., 2022), AWQ (Lin et al., 2024a), and OmniQuant (Shao et al., 2023) enable 4-bit weight representation with minimal performance degradation. Knowledge distillation approaches like MiniLLM (Gu et al., 2023) and GKD (Agarwal et al., 2024) transfer knowledge from larger teacher models to smaller students, with the latter introducing on-policy distillation to address train-inference distribution mismatch. Combined approaches leveraging pruning with distillation have yielded results comparable to the strongest baselines (Sreenivas et al., 2024; Muralidharan et al., 2024).

Building upon these structural approaches, unstructured pruning enhances LLM compression by flexibly removing weights, often achieving higher accuracy at high sparsity levels compared to structured methods (Lee et al., 2024). Recent advancements in this field include post-training one-shot pruning techniques such as SparseGPT (Frantar & Alistarh, 2023), which leverages second-order approximations to prune over 50% of weights in a 175B model while maintaining performance with minimal loss, and Wanda (Sun et al., 2023), which simplifies the pruning process through activation-aware magnitude pruning for efficient sparsity. Other notable contributions in this category, such as OWL (Yin et al., 2023), Tyr-the-Pruner (Li et al., 2025), and GBLM-Pruner (Das et al., 2023), further refine sparsity allocation and incorporate gradient correction to enhance pruning outcomes. Additionally, optimization-based methods like ALPS (Meng et al., 2024) approach pruning as a constrained optimization problem, while DSnoT (Zhang et al., 2023) iteratively refines weights to optimize results over multiple steps.

Beyond pruning techniques, model quantization is essential for deploying LLMs efficiently, reducing memory and computation by lowering the precision of weights and activations (Gong et al., 2024). However, the 'outlier' problem, where a few large values dominate the quantization range, poses a significant challenge (Ashkboos et al., 2024b). Early solutions like LLM.int8() (Dettmers et al., 2022) addressed this by retaining some activations in higher precision. Recent advancements include GPTQ (Frantar et al., 2022), which uses second-order information for accurate 4-bit weight quantization, and AWQ (Lin et al., 2024a), which protects key weights based on activation statistics. For activation quantization, SmoothQuant (Xiao et al., 2023) redistributes outliers to enable 8-bit quantization, while QuaRot (Ashkboos et al., 2024b) and QuIP (Chee et al., 2023) employ rotation transformations for 4-bit and 2-bit quantization, respectively. Additionally, QUIK (Ashkboos et al., 2023) and QServe (Lin et al., 2024b) combine quantization with system optimizations for practical deployment. Other notable methods include PrefixQuant (Chen et al., 2024a) and MergeQuant (Wang et al., 2025) for efficient static quantization.

With models compressed through the aforementioned techniques, large language model inference faces significant computational challenges that researchers address through various optimization approaches. Recent work has focused on KV cache management techniques such as Page-

dAttention (Kwon et al., 2023), which treats cache as virtual memory to reduce fragmentation, SpeCache (Jie et al., 2025), which intelligently prefetches needed keys, and QuaRot (Ashkboos et al., 2024b) for outlier-free 4-bit inference. Algorithmic optimizations like Cascade Speculative Drafting (Chen et al., 2024b) leverage a tiered approach where smaller models draft for larger ones, while N-gram masked self-attention Chelba et al. (2020) truncates attention windows. Memory efficiency improvements include SWAT (Fu et al., 2025) for sliding window attention and Inf-MLLM (Ning et al., 2024) for streaming inference via attention saddle patterns. Architecture innovations explore MoE approaches like Read-ME (Cai et al., 2024) which refactors dense models into router-decoupled experts, and Dynamic-LLaVA (Huang et al., 2024) which sparsifies vision-language contexts. System-level optimizations include operator fusion in Faster-Transformer (Aminabadi et al., 2022), FlashAttention's IO-aware computation (Dao et al., 2022), FlashDecoding++ (Hong et al., 2024) with asynchronous softmax, and specialized schedulers like Sarathi-Serve (Agrawal et al., 2024), vLLM (Kwon et al., 2023), and Llumnix (Sun et al., 2024) that balance throughput and latency through innovative resource management strategies.

## APPENDIX A.1: MOTIVATION

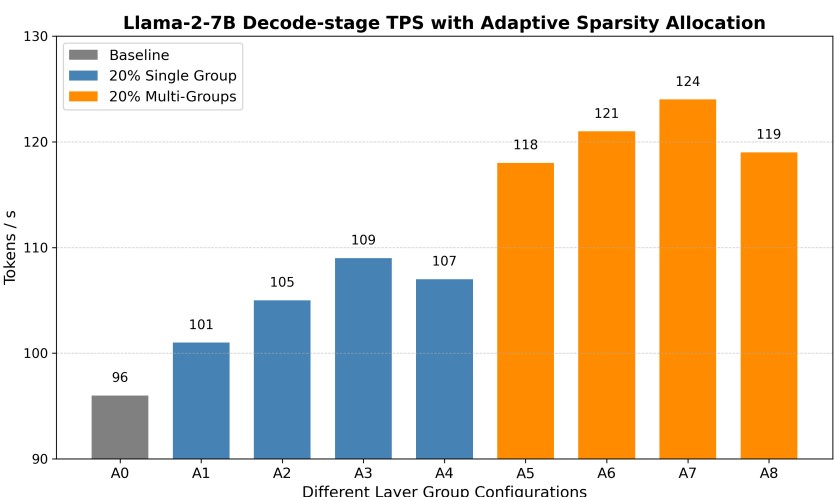

Figure 1: Decode-stage TPS with sparsity allocation across layer groups, based on 8-bit QuaRot quantization. Task and hardware: left-to-right language-model generation, each run takes a 2048-token prompt from the WikiText-103 validation set, allowing Llama-2-7B to autoregressively generate the next 256 tokens, and runs on a single NVIDIA RTX 3090 GPU. Variants on the X-axis denote different layer group sparsity configurations: A0: baseline, no sparsity applied (96 tokens/s); A1: 20% sparsity applied to layers 1-8 (early attention layers, 101 tokens/s); A2: 20% sparsity applied to layers 9-16 (middle attention layers, 105 tokens/s); A3: 20% sparsity applied to layers 17-24 (middle-late layers, 109 tokens/s); A4: 20% sparsity applied to layers 25-32 (late transformer layers, 107 tokens/s); A5: 20% sparsity applied to layers 1-8 and 17-24 simultaneously (118 tokens/s); A6: 20% sparsity applied to layers 9-16 and 25-32 simultaneously (121 tokens/s); A7: 20% sparsity applied to layers 1-8, 9-16, and 17-24 (124 tokens/s); A8: 20% sparsity applied to all layer groups 1-32 (119 tokens/s). Y-axis: measured tokens-per-second during decode phase. Higher is better.

In the deployment of LLMs, quantization and pruning have evolved as independent acceleration techniques with minimal integration. In our explorations, we conduct a series of attempts to investigate the combination of both techniques. Figure 1 demonstrates the inference acceleration achieved on the Llama-2-7B model in a language generation task, after applying 20% sparsity rate to various layers and layer combinations. We observe a significant improvement in model inference speed. Figure 2 shows the model's performance across different quantization and sparsification configurations. As the bit-width of quantization decreases and the sparsity ratio increases, the perplexity rises substantially, indicating a clear trade-off between efficiency and performance.

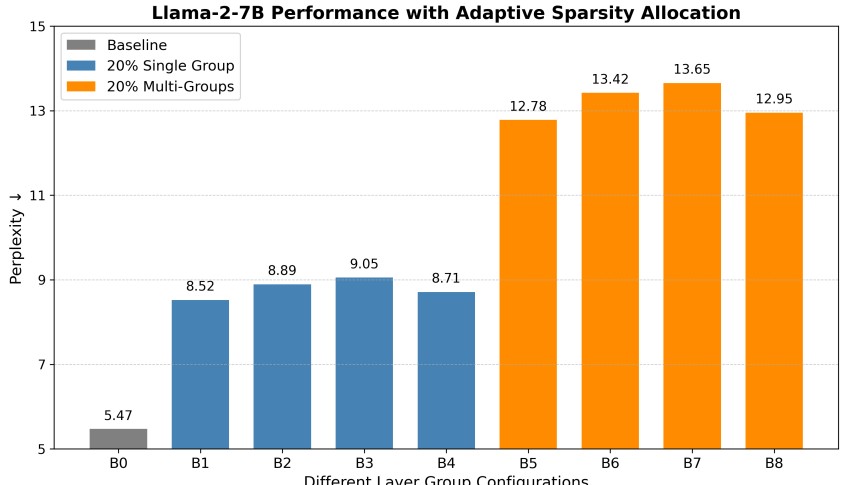

Figure 2: Performance of quantization and layer-group sparsity allocation on Llama-2-7B. Task and hardware: left-to-right language-model evaluation on the WikiText-2 validation subset. The evaluation uses approximately 1 000 passages, each with a sequence length of 2048, running on a single NVIDIA RTX 3090 GPU with 8-bit QuaRot quantization applied to all weights. Variants on the X-axis denote different layer group sparsity configurations: B0: 8-bit quantized baseline, no additional sparsity (5.47 perplexity); B1: 8-bit quantization + 20% sparsity on layers 1-8 (8.52 perplexity, 55.8% degradation); B2: 8-bit quantization + 20% sparsity on layers 9-16 (8.89 perplexity, 62.5% degradation); B3: 8-bit quantization + 20% sparsity on layers 17-24 (9.05 perplexity, 65.4% degradation); B4: 8-bit quantization + 20% sparsity on layers 25-32 (8.71 perplexity, 59.2% degradation); B5: 8-bit quantization + 20% sparsity on layers 1-8 and 17-24 (12.78 perplexity, 133.5% degradation); B6: 8-bit quantization + 20% sparsity on layers 9-16 and 25-32 (13.42 perplexity, 145.2% degradation); B7: 8-bit quantization + 20% sparsity on layers 1-8, 9-16, and 17-24 (13.65 perplexity, 149.5% degradation); B8: 8-bit quantization + 20% sparsity on all layer groups (12.95 perplexity, 136.7% degradation). Y-axis: measured perplexity on WikiText-2 (↓). Lower is better.

### A.1.1 Layer-Group Sensitivity Analysis

The experimental results reveal distinct sensitivity patterns across different layer groups in the Llama-2-7B architecture under 20% sparsity allocation. Early layers (1-8) demonstrate moderate resilience to sparsification, with configuration A1 achieving 5.2% throughput improvement while B1 exhibits 55.8% perplexity degradation. This substantial degradation, even in early layers, highlights the critical nature of all computational pathways in modern LLMs.

Middle layers show varying sensitivity patterns: layers 9-16 (A2/B2) achieve 9.4% throughput gains with 62.5% quality degradation, while layers 17-24 (A3/B3) demonstrate the highest single-group acceleration of 13.5% with 65.4% perplexity increase. These middle-late layers exhibit the most severe single-group degradation, suggesting their critical role in semantic processing.

Late layers (25-32) show a balanced pattern with A4 achieving 11.5% throughput improvement while B4 incurs 59.2% quality loss. This pattern suggests that while late layers contain exploitable computational redundancy, their sparsification significantly impacts final output quality, though less severely than middle-late layers.

### A.1.2 Multi-Group Sensitivity Analysis

The multi-group configurations (A5-A8, B5-B8) reveal the catastrophic nature of naive sparsity combination strategies. Configuration A5, combining layers 1-8 and 17-24 with 20% sparsity each, achieves 22.9% throughput improvement but incurs 133.5% perplexity degradation (B5). This represents a fundamental phase transition where the cumulative effect far exceeds the sum of individual impacts.

The most aggressive multi-group configuration A7 (layers 1-8, 9-16, 17-24) demonstrates maximum throughput gains of 29.2% but suffers catastrophic quality collapse with 149.5% perplexity degradation (B7). Paradoxically, the all-groups configuration A8 shows slightly reduced acceleration (24.0%) and degradation (136.7%), suggesting complex non-linear interactions when the entire model undergoes simultaneous sparsification.

## APPENDIX A.2: QUANTIZATION FRAMEWORK

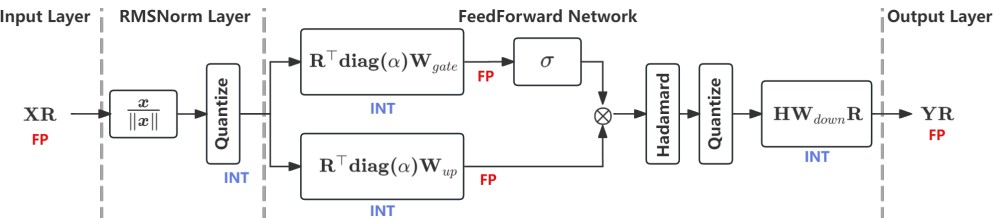

Figure 3: Low-bit quantization pipeline for a feed-forward network block with quantization and rotation.

In the framework illustrated in Figure 3, the hidden state $X$ is first multiplied by a random orthogonal matrix $R$ stored in FP($m$) precision. Because $R^\top R = I$, this transformation *preserves Euclidean norms and inner products*, i.e. $(RX)^\top(RY) = X^\top Y$ (Ashkboos et al., 2024b). Consequently, substituting $X \mapsto RX$ and $W \mapsto R^\top W$ leaves each linear layer's output unchanged:

$$\underbrace{XW}_{\text{original}} = \underbrace{(RX)(R^\top W)}_{\text{rotated}}. \tag{1}$$

This computational invariance (Ashkboos et al., 2024a) enables the framework to redistribute heavy-tailed activation energy across dimensions before quantization, thereby mitigating the "outlier" problem that hampers uniform low-bit encodings (Ashkboos et al., 2024b).

After rotation, the activations are quantized to INT($n$). The feed-forward weights $W_{\text{gate}}$ and $W_{\text{up}}$ are premultiplied by $R^\top$ and rescaled by $\text{diag}(\alpha)$ (RMSNorm) (Zhang & Sennrich, 2019b). The non-linearity $\sigma$ is applied, followed by an on-the-fly Hadamard transform $H$; this matrix is also orthogonal, so fusing $H$ into the down-projection $W_{\text{down}}$ preserves functional equivalence while further flattening variance. A second INT($n$) quantization converts the output back to low precision before casting to FP($m$).

Because every orthogonal transform is absorbed into adjacent linear layers during an *offline* pre-processing step, the run-time kernel sequence matches that of the baseline network while operating entirely on 4-bit integers. Combined with per-channel GPTQ calibration (Frantar et al., 2022), this rotation–quantize–fuse pipeline achieves end-to-end INT4 inference without mixed-precision fallbacks, all while guaranteeing mathematical equivalence to the original FP16 model.

## APPENDIX A.3: DYNAMIC PROGRAMMING BACKGROUND

### A.3.1 ALGORITHMIC PARADIGM

Dynamic programming is both a mathematical optimization method and an algorithmic paradigm developed by Richard Bellman in the 1950s (Bellman, 1957). The approach simplifies complex problems by breaking them down into simpler subproblems in a recursive manner, then combining their solutions to solve the original problem (Cormen et al., 2009).

The paradigm applies when a problem exhibits two fundamental properties: optimal substructure and overlapping subproblems. Unlike divide-and-conquer algorithms, which solve entirely independent subproblems, dynamic programming exploits the fact that subproblems are not independent, the same subproblems arise repeatedly during the recursive solution process.

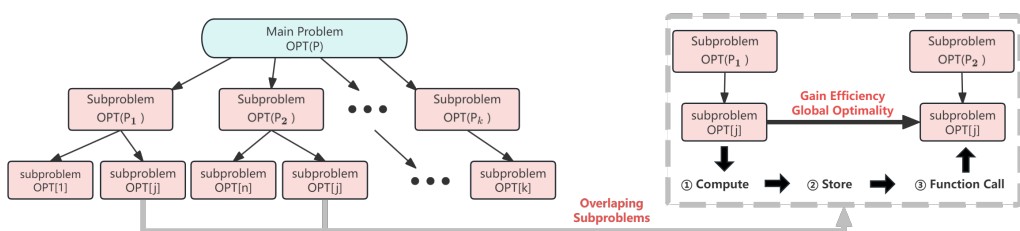

Figure 4: Illustration of dynamic programming approach showing problem decomposition into sub-problems.

The essence of dynamic programming lies in avoiding redundant computation by storing solutions to subproblems for later reuse. This memoization strategy transforms algorithms with exponential time complexity into polynomial-time solutions (Dasgupta et al., 2006).

### A.3.2 OPTIMAL SUBSTRUCTURE PROPERTY

A problem exhibits optimal substructure if an optimal solution can be constructed efficiently from optimal solutions of its subproblems (Cormen et al., 2009). This property is fundamental because it ensures that solving subproblems optimally contributes to the global optimum.

Mathematically, optimal substructure can be expressed as a functional relationship. For a problem $P$, if $\text{OPT}(P)$ represents the optimal solution, then:

$$\text{OPT}(P) = f(\text{OPT}(P_1), \text{OPT}(P_2), \ldots, \text{OPT}(P_k)), \tag{2}$$

where $f$ is a function that combines optimal solutions of subproblems $\{P_i\}$ to yield the optimal solution of the original problem.

The canonical example demonstrating optimal substructure is the shortest path problem. If the shortest path from vertex $u$ to vertex $v$ passes through intermediate vertex $w$, then this path must consist of the shortest path from $u$ to $w$ concatenated with the shortest path from $w$ to $v$. Any deviation from this principle would contradict the optimality of the overall path (Cormen et al., 2009).

Verification of optimal substructure typically employs proof by contradiction. If a subproblem within an alleged optimal solution were not itself optimal, then substituting the actual optimal sub-solution would improve the overall solution, contradicting the assumption of optimality.

### A.3.3 OVERLAPPING SUBPROBLEMS PROPERTY

The overlapping subproblems property distinguishes dynamic programming from divide-and-conquer approaches. This property requires that the space of subproblems be small relative to the total number of recursive calls, meaning the same subproblems are solved repeatedly (Cormen et al., 2009).

In problems with overlapping subproblems, naive recursive implementations typically exhibit exponential time complexity due to redundant calculations. The Fibonacci sequence computation illustrates this phenomenon: calculating $F(n)$ requires computing $F(n-1)$ and $F(n-2)$, where $F(n-1)$ itself requires $F(n-2)$ and $F(n-3)$, leading to multiple evaluations of the same Fibonacci numbers.

The mathematical characterization of this property can be expressed through the recurrence tree structure. If $T(n)$ represents the number of subproblems of size $n$ solved during the recursive process, and $S(n)$ represents the number of distinct subproblems of size $n$, then overlapping subproblems exist when $T(n) >> S(n)$ for sufficiently large $n$.

Dynamic programming exploits this overlap through memoization, storing computed solutions in a table for subsequent lookup. This technique reduces the time complexity from exponential to polynomial by ensuring each distinct subproblem is solved exactly once (Dasgupta et al., 2006).

### A.3.4 Mathematical Formulation

Dynamic programming algorithms follow a general mathematical structure based on the principle of optimality. The fundamental recurrence relation takes the form:

$$\text{OPT}[i] = \min_{j \in \mathcal{J}(i)} \left\{ \text{OPT}[j] + \text{Cost}(j, i) \right\}, \tag{3}$$

where $\text{OPT}[i]$ represents the optimal solution value for subproblem $i$, $\mathcal{J}(i)$ denotes the set of feasible predecessor states, and $\text{Cost}(j, i)$ represents the immediate cost of transitioning from state $j$ to state $i$.

The state space design constitutes a critical component of dynamic programming formulations. States must encapsulate sufficient information to make optimal decisions while maintaining computational tractability. The dimensionality of the state space directly affects both algorithm correctness and efficiency.

Boundary conditions provide the foundation for recursive computations. These base cases correspond to trivial subproblems that can be solved directly without further decomposition:

$$\text{OPT}[0] = \text{base\_value}. \tag{4}$$

The optimization objective varies depending on the problem context. Minimization problems seek $\min_j$, maximization problems use $\max_j$, and counting problems sum over all valid transitions. Each formulation requires careful consideration of how subproblem solutions combine.

### A.3.5 Implementation Strategies

Dynamic programming algorithms can be implemented using two primary approaches: top-down memoization and bottom-up tabulation (Cormen et al., 2009).

Top-down memoization preserves the natural recursive structure while avoiding redundant computations through result caching. The algorithm begins with the original problem and recursively decomposes it into subproblems, storing computed results in a memoization table:

$$\text{memo}[i] = \begin{cases} \text{computed\_value} & \text{if already solved,} \\ \text{recursive\_solve}(i) & \text{otherwise.} \end{cases} \tag{5}$$

Bottom-up tabulation systematically solves subproblems in order of increasing size, building solutions iteratively from base cases to the final answer. This approach typically follows the pattern:

$$\text{for } i = 1 \text{ to } n: \quad \text{OPT}[i] = f(\text{OPT}[0], \dots, \text{OPT}[i-1]). \tag{6}$$

The choice between these approaches depends on several factors. Memoization proves advantageous when only a subset of all possible subproblems requires solution, while tabulation offers better cache locality and predictable memory access patterns for dense subproblem spaces.

Space optimization techniques can significantly reduce memory requirements. Common optimizations include maintaining only the most recently computed results when the recurrence relation depends on a fixed number of previous values, reducing space complexity from $O(n)$ to $O(1)$ in many cases.

## Appendix A.4: ASAF Framework Visual Illustration

### A.4.1 Coarse-Grained Optimization Phase

Figure 5 illustrates the coarse-grained optimization phase of the ASAF framework, which addresses the fundamental challenge of determining optimal layer grouping strategies and sparsity interval refinement. The process begins with individual transformer layers $L_1, L_2, \dots, L_p$ from the large language model, where each layer is represented as an independent computational unit with specific FLOPs characteristics $\phi_l$.

The grouping transformation process aggregates these individual layers into cohesive groups $G_1, G_2, \dots, G_n$, where each group $G_i$ contains a set of consecutive layers $\mathcal{L}_i = \{l_{\text{start}}^{(i)}, l_{\text{start}}^{(i)} +$

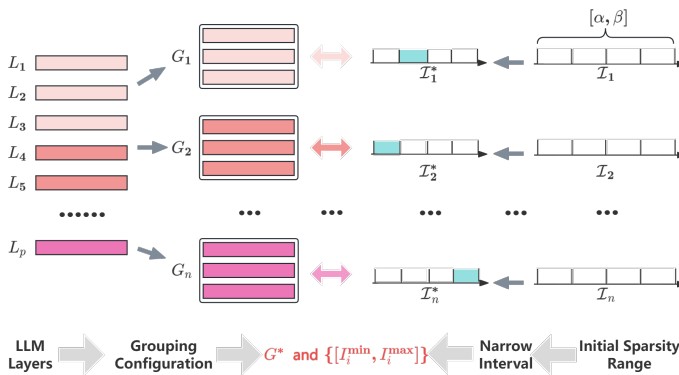

Figure 5: Coarse-grained optimization process in the ASAF framework. The phase transforms individual LLM layers $L_1, L_2, \ldots, L_p$ into optimal grouping configurations $G_1, G_2, \ldots, G_n$ while simultaneously refining sparsity search intervals from the initial range $[\alpha, \beta]$ to narrowed intervals $\{[I_i^{\min}, I_i^{\max}]\}$ for each group, where $G^* = n$ denotes the optimal number of groups. The optimization process employs iterative interval refinement strategies $\mathcal{I}_1^*, \mathcal{I}_2^*, \ldots, \mathcal{I}_n^*$ to achieve computational efficiency while maintaining sensitivity constraints.

$1, \ldots, l_{\mathrm{end}}^{(i)}\}$. This grouping strategy ensures that layers within each group can be managed collectively while preserving the sequential structure of the transformer architecture.

The simultaneous sparsity interval refinement process transforms the initial broad search space $[\alpha, \beta]$ into narrowed, group-specific intervals $\{[I_i^{\min}, I_i^{\max}]\}$. This refinement mechanism employs dynamic programming principles to systematically reduce the search complexity while maintaining optimality guarantees. The refinement process utilizes the RefineIntervals($\cdot$) and NarrowInterval($\cdot$) functions to achieve progressive convergence toward optimal sparsity allocations.

The optimization trajectory shown in the figure demonstrates how the algorithm iteratively narrows the search space through successive refinements $\mathcal{I}_1^*, \mathcal{I}_2^*, \ldots, \mathcal{I}_n^*$, where each iteration produces tighter bounds on feasible sparsity rates. This process continues until convergence criteria are satisfied, yielding the optimal number of groups $G^*$ and their corresponding refined sparsity intervals.

### A.4.2 FINE-GRAINED OPTIMIZATION PHASE

Figure 6 depicts the fine-grained optimization phase, which performs precise decision-making within the framework established by the coarse-grained phase. This phase receives as input the optimal number of groups $G^*$ and the refined sparsity intervals $\{[I_i^{\min}, I_i^{\max}]\}$ determined in the previous phase.

The layer allocation component systematically determines the exact number of consecutive layers assigned to each group. This process involves solving the optimization problem $\{j^*, s^*\}_i = \arg\min_{j,s}\{T[i][j][s]\}$ for each group $i$, where $T[i][j][s]$ represents the total FLOPs cost for group $i$ containing $j$ consecutive layers with sparsity rate $s$. The allocation process ensures complete coverage of all transformer layers while respecting continuity constraints.

The sparsity selection component operates in parallel with layer allocation, determining optimal sparsity rates from the discretized candidate sets generated by the Discretize($\cdot$) function. For each group $i$, the algorithm evaluates sparsity candidates $S_{\mathrm{cand}} = \{I_i^{\min} + k\Delta : k \in \mathbb{Z}, 0 \leq k \leq \lfloor(I_i^{\max} - I_i^{\min})/\Delta\rfloor\}$ with sampling resolution $\Delta = 0.5\%$.

The progressive convergence mechanism illustrated in the figure shows how the optimization iteratively refines both layer assignments and sparsity selections. The process employs tabulation table lookups $H[l][j][s]$ to efficiently evaluate different configurations without redundant computation. Each iteration improves upon previous solutions by leveraging the optimal substructure property inherent in the problem formulation.

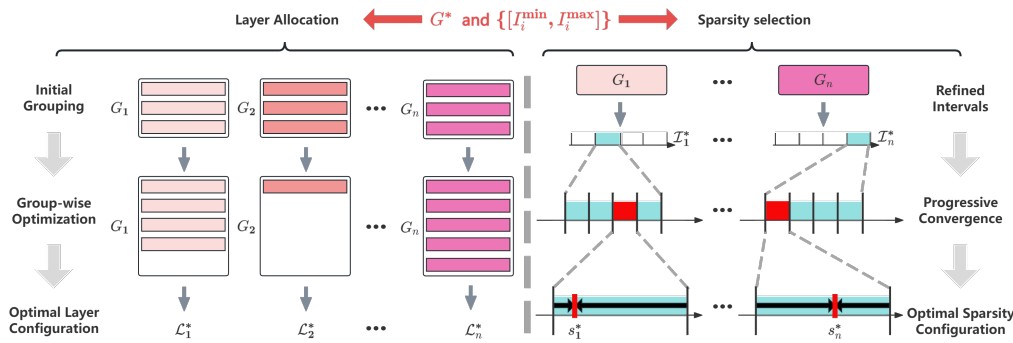

Figure 6: Fine-grained optimization process in the ASAF framework. Building upon the coarse-grained results $G^*$ and refined intervals $\{[I_i^{\min}, I_i^{\max}]\}$, this phase performs precise layer allocation and sparsity selection through group-wise optimization. The process progressively converges from initial grouping configurations through iterative optimization to optimal layer configurations $\mathcal{L}_1^*, \mathcal{L}_2^*, \ldots, \mathcal{L}_n^*$ and optimal sparsity rates $s_1^*, s_2^*, \ldots, s_n^*$, where $G^* = n$ denotes the optimal number of groups.

The final optimization output produces the complete ASAF solution: optimal layer configurations $\mathcal{L}_1^*, \mathcal{L}_2^*, \ldots, \mathcal{L}_n^*$ specifying the exact layer ranges for each group, and optimal sparsity rates $s_1^*, s_2^*, \ldots, s_n^*$ that minimize total computational FLOPs while satisfying sensitivity constraints $\sum_{i=1}^{G^*} \xi(\mathcal{L}_i^*, s_i^*) \leq \delta_{\max}$.

The two-phase decomposition strategy demonstrated in Figures 5 and 6 exemplifies the dynamic programming approach by transforming a complex combinatorial optimization problem into a sequence of manageable subproblems. This decomposition achieves polynomial time complexity while maintaining optimality guarantees, making the ASAF framework computationally tractable for practical large language model deployment scenarios.

## APPENDIX A.5: COARSE-GRAINED OPTIMIZATION

The coarse-grained optimization phase serves as the foundation of our ASAF framework by establishing the optimal group structure and narrowing the sparsity search space for subsequent fine-grained optimization. This phase addresses the fundamental challenge of determining how many layer groups should be formed and what sparsity intervals each group should operate within, effectively decomposing the exponential search space into manageable subproblems.

### A.5.1 MATHEMATICAL FORMULATION

The coarse-grained optimization addresses the first subproblem of our original formulation in Equation 3 by minimizing the total computational FLOPs across all possible group configurations while satisfying accuracy degradation constraints:

$$\{G^*, \{I_i^*\}_{i=1}^{G^*}\} = \arg\min_{G, \{I_i\}_{i=1}^{G}} \left\{ \sum_{i=1}^{G} \min_{\mathcal{L}_i, s_i \in I_i} \sum_{l \in \mathcal{L}_i} \phi_l \times (1 - s_i) \right\}, \tag{7}$$

where $G$ represents the number of groups, $I_i$ denotes the sparsity interval for group $i$, $\mathcal{L}_i$ represents consecutive layers in group $i$, $s_i$ is the sparsity rate for group $i$, and $\phi_l$ is the original computational cost of layer $l$. The constraint ensures $\sum_{i=1}^{G} \xi(\mathcal{L}_i, s_i) \leq \delta_{\max}$, where $\xi(\mathcal{L}_i, s_i)$ quantifies the accuracy degradation caused by applying sparsity rate $s_i$ to layer group $\mathcal{L}_i$.

The dynamic programming state formulation defines $\mathrm{DP}_{\mathrm{coarse}}[g][b]$ as the minimum total FLOP cost when using exactly $g$ groups with discretized accuracy degradation budget $b$:

$$\mathrm{DP}_{\mathrm{coarse}}[g][b] = \min_{\{I_i\}_{i=1}^g} \left\{ \sum_{i=1}^{g} \min_{\mathcal{L}_i, s_i \in I_i} \sum_{l \in \mathcal{L}_i} \phi_l \times (1 - s_i) \right\}, \tag{8}$$

where the state represents the optimal cost achievable with $g$ groups under accuracy degradation budget $b \times \Delta$ (with $\Delta$ being the discretization step), and the minimization considers all valid interval partitions and corresponding layer-sparsity assignments.

The state transition equation considers all possible ways to add one more group by selecting an appropriate interval subset:

$$\mathrm{DP}_{\mathrm{coarse}}[g][b] = \min_{I \subseteq [\alpha, \beta]} \left\{ \mathrm{OptimalCost}(I) + \mathrm{DP}_{\mathrm{coarse}}[g-1][b - \xi_{\mathrm{cost}}(I)/\Delta] \right\}, \tag{9}$$

where $I$ represents the sparsity interval assigned to the $g$-th group, $\mathrm{OptimalCost}(I)$ denotes the minimum FLOP cost achievable within interval $I$, and $\xi_{\mathrm{cost}}(I)$ represents the accuracy degradation.

---

**Algorithm 1** Coarse-grained Optimization

---

**Require:** Layer count $p$, sparsity range $[\alpha, \beta]$, accuracy degradation threshold $\delta_{\mathrm{max}}$
**Ensure:** Optimal group number $G^*$ and intervals $\{[I_i^{\mathrm{min}}, I_i^{\mathrm{max}}]\}$
 1: Initialize $\mathrm{DP}_{\mathrm{coarse}}[0..p][0..\delta_{\mathrm{max}}/\Delta] \leftarrow \infty$
 2: Set boundary condition: $\mathrm{DP}_{\mathrm{coarse}}[0][0] \leftarrow 0$
 3: Generate interval candidates $\mathcal{I} = \{I : I \subseteq [\alpha, \beta]\}$
 4: **Main DP Loop:**
 5: **for** $g = 1$ **to** $p$ **do**
 6:    **for** $b = 0$ **to** $\delta_{\mathrm{max}}/\Delta$ **do**
 7:       **for** each interval $I \in \mathcal{I}$ **do**
 8:          cost $\leftarrow \mathrm{OptimalCost}(I)$ using tabulation $H$
 9:          accuracy_cost $\leftarrow \xi_{\mathrm{cost}}(I)/\Delta$ using tabulation $\Xi$
10:          **if** $b - $ accuracy_cost $\geq 0$ **then**
11:             total_cost $\leftarrow$ cost $+ \mathrm{DP}_{\mathrm{coarse}}[g-1][b - $ accuracy_cost$]$
12:             $\mathrm{DP}_{\mathrm{coarse}}[g][b] \leftarrow \min(\mathrm{DP}_{\mathrm{coarse}}[g][b], \text{total\_cost})$
13:          **end if**
14:       **end for**
15:    **end for**
16: **end for**
17: **Solution Extraction:**
18: $G^* \leftarrow \arg\min_g \{\min_b \mathrm{DP}_{\mathrm{coarse}}[g][b]\}$
19: $\{I_i^{\mathrm{min}}, I_i^{\mathrm{max}}\} \leftarrow \mathrm{BacktrackOptimalIntervals}(\mathrm{DP}_{\mathrm{coarse}}, G^*)$
20: **return** $G^*$, $\{[I_i^{\mathrm{min}}, I_i^{\mathrm{max}}]\}_{i=1}^{G^*}$ =0

---

### A.5.2 ALGORITHMIC ANALYSIS

The dynamic programming formulation operates on a two-dimensional state space where $\mathrm{DP}_{\mathrm{coarse}}[g][b]$ represents the minimum achievable computational cost when utilizing exactly $g$ groups with discretized accuracy degradation budget $b$. The discretization parameter $\Delta = 0.5\%$ transforms the continuous constraint into a tractable discrete optimization problem while maintaining sufficient precision for practical deployment.

The initialization establishes boundary conditions essential for correctness. The infinite initialization for $\mathrm{DP}_{\mathrm{coarse}}[g][b]$ ensures only valid transitions result in finite costs, while $\mathrm{DP}_{\mathrm{coarse}}[0][0] = 0$ represents the base case with no groups and no degradation.

The interval generation constructs candidate sparsity intervals $\mathcal{I}$ that partition $[\alpha, \beta]$ into meaningful subranges. Each interval $I \in \mathcal{I}$ represents a potential sparsity operating range for a layer group, enabling exploration of different sparsity allocation granularities.

The core dynamic programming loop systematically explores group counts and accuracy budgets through three nested iterations. The outermost loop over $g$ considers configurations from single

groups to maximum granularity where $G_{\max} = p$. The middle loop over budget $b$ discretizes the accuracy constraint, while the innermost loop over intervals $I$ evaluates each potential sparsity assignment.

Within each iteration, the algorithm queries tabulation tables $H$ and $\Xi$ for precomputed costs and degradation values. The tabulation mechanism transforms expensive evaluations into constant-time operations, enabling scalability to large layer counts. The cost computation combines immediate interval contribution with optimal subproblem costs, maintaining optimal substructure.

The feasibility check $b - \text{accuracy\_cost} \geq 0$ ensures configurations remain within the accuracy threshold $\delta_{\max}$. When feasible configurations are identified, the minimization operation updates the dynamic programming table only for improved solutions.

The solution extraction identifies the optimal configuration by examining all computed states: $G^* = \arg\min_g \{\min_b \text{DP}_{\text{coarse}}[g][b]\}$. The backtracking procedure reconstructs specific interval assignments by tracing decisions that led to the optimal solution, producing refined sparsity intervals that narrow the search space from $[\alpha, \beta]$.

**State Transition Intuition.** The dynamic programming transitions embody intuitive decision-making processes. In the coarse-grained phase, each state transition addresses the question: "Given that I have optimally allocated the first $g-1$ groups, what sparsity interval should I assign to the $g$-th group to minimize total cost while staying within the accuracy budget?" This decomposition enables systematic exploration of interval assignments without reconsidering previous group decisions. The state $\text{DP}_{\text{coarse}}[g][b]$ encapsulates all optimal ways to use exactly $g$ groups with budget $b$, allowing efficient evaluation of adding one more group with a specific interval choice.

### A.5.3 COMPLEXITY ANALYSIS

Upon completion, the coarse-grained optimization produces two critical outputs: the optimal number of groups $G^*$ that minimizes computational overhead while satisfying accuracy constraints, and refined sparsity intervals $\{[I_i^{\min}, I_i^{\max}]\}_{i=1}^{G^*}$ that narrow the search space from the original range $[\alpha, \beta]$. These outputs provide essential guidance for the subsequent fine-grained optimization phase, significantly reducing the search space while preserving the potential for globally optimal solutions.

## APPENDIX A.6: FINE-GRAINED OPTIMIZATION

The fine-grained optimization phase receives the optimal group number $G^*$ and refined sparsity intervals from the coarse-grained phase and determines the precise layer allocation and sparsity rate assignment that minimizes computational FLOPs while satisfying accuracy constraints. This phase operates on a significantly reduced search space compared to the original problem, enabling detailed optimization within the refined parameter ranges.

### A.6.1 MATHEMATICAL FORMULATION

The fine-grained optimization addresses the second subproblem of our original formulation by minimizing the exact total computational FLOPs through optimal layer partitioning and sparsity assignment:

$$\{\{\mathcal{L}_i^*\}_{i=1}^{G^*}, \{s_i^*\}_{i=1}^{G^*}\} = \arg \min_{\{\mathcal{L}_i\}_{i=1}^{G^*}, \{s_i\}_{i=1}^{G^*}} \left\{ \sum_{i=1}^{G^*} \sum_{l \in \mathcal{L}_i} \phi_l \times (1 - s_i) \right\}, \tag{10}$$

where $\{\mathcal{L}_i\}_{i=1}^{G^*}$ represents the layer allocation with each $\mathcal{L}_i$ containing consecutive layers, and $\{s_i\}_{i=1}^{G^*}$ denotes the sparsity rates. The constraints ensure $\sum_{i=1}^{G^*} \xi(\mathcal{L}_i, s_i) \leq \delta_{\max}$, $s_i \in [I_i^{\min}, I_i^{\max}]$, $\mathcal{L}_i$ are consecutive, $\bigcup_{i=1}^{G^*} \mathcal{L}_i = \{1, 2, \ldots, p\}$, and $\mathcal{L}_i \cap \mathcal{L}_j = \emptyset$ for $i \neq j$.

The dynamic programming state formulation defines $\text{DP}_{\text{fine}}[i][g][b]$ as the minimum total FLOP cost for optimally partitioning layers $[i..p]$ into exactly $g$ consecutive groups with remaining accuracy

degradation budget $b$:

$$\text{DP}_{\text{fine}}[i][g][b] = \min_{\{\mathcal{L}_k\}_{k=1}^g, \{s_k\}_{k=1}^g} \left\{ \sum_{k=1}^g \sum_{l \in \mathcal{L}_k} \phi_l \times (1 - s_k) \right\}, \quad (11)$$

where the state covers layers from position $i$ to $p$, requires exactly $g$ groups, has remaining budget $b$, and ensures $\bigcup_{k=1}^g \mathcal{L}_k = \{i, i+1, \ldots, p\}$ with each $\mathcal{L}_k$ consecutive, $s_k \in [I_k^{\min}, I_k^{\max}]$, and $\sum_{k=1}^g \xi(\mathcal{L}_k, s_k) \leq b$.

---

**Algorithm 2** Fine-grained Optimization

---

**Require:** $G^*$ groups, intervals $\{[I_i^{\min}, I_i^{\max}]\}$, tabulation tables $H, \Xi$
**Ensure:** Optimal allocation $\{\mathcal{L}_i^*\}$ and sparsity rates $\{s_i^*\}$
1: Initialize $\text{DP}_{\text{fine}}[1..p+1][0..G^*][0..\delta_{\max}/\Delta] \leftarrow \infty$
2: Initialize $\text{choice}[1..p+1][0..G^*][0..\delta_{\max}/\Delta] \leftarrow \emptyset$
3: Set boundary condition: $\text{DP}_{\text{fine}}[p+1][0][b] \leftarrow 0$ for all $b \geq 0$
4: **Backward DP Construction:**
5: **for** $i = p$ **down to** $1$ **do**
6:     **for** $g = 1$ **to** $\min(G^*, p - i + 1)$ **do**
7:         **for** $b = 0$ **to** $\delta_{\max}/\Delta$ **do**
8:             **for** $j = i$ **to** $p - g + 1$ **do**
9:                 **for** $s \in \text{Discretize}([I_{G^*-g+1}^{\min}, I_{G^*-g+1}^{\max}])$ **do**
10:                     group_cost, accuracy_cost $\leftarrow H[i][j - i + 1][s], \Xi[i][j - i + 1][s]/\Delta$
11:                     **if** $b - $ accuracy_cost $\geq 0$ **then**
12:                         total_cost $\leftarrow$ group_cost $+ \text{DP}_{\text{fine}}[j+1][g-1][b - \text{accuracy\_cost}]$
13:                         **if** total_cost $< \text{DP}_{\text{fine}}[i][g][b]$ **then**
14:                             $\text{DP}_{\text{fine}}[i][g][b] \leftarrow$ total_cost
15:                             $\text{choice}[i][g][b] \leftarrow (j, s)$
16:                       **end if**
17:                   **end if**
18:                 **end for**
19:             **end for**
20:         **end for**
21:     **end for**
22: **end for**
23: **Solution Reconstruction:**
24: $\{\mathcal{L}_i^*, s_i^*\} \leftarrow \text{BacktrackSolution}(\text{choice}, 1, G^*, \delta_{\max}/\Delta)$
25: **return** $\{\mathcal{L}_i^*\}_{i=1}^{G^*}, \{s_i^*\}_{i=1}^{G^*} = 0$

---

The state transition equation jointly enumerates all possible first-group formations and sparsity assignments within the corresponding refined interval. The transition first identifies the optimal group end position $j$ and sparsity rate $s$:

$$j^*, s^* = \arg \min_{\substack{j \in [i, p-g+1] \\ s \in [I_{G^*-g+1}^{\min}, I_{G^*-g+1}^{\max}]}} \{H[i][j - i + 1][s] + \Xi[i][j - i + 1][s]\}. \quad (12)$$

Then the state transition equation becomes:

$$\text{DP}_{\text{fine}}[i][g][b] = H[i][j^* - i + 1][s^*] + \text{DP}_{\text{fine}}[j^* + 1][g - 1][b - \Xi[i][j^* - i + 1][s^*]], \quad (13)$$

where $j$ defines the end of the first group (layers $i$ to $j$), $s$ is the sparsity rate from the refined interval for this group position, $H[i][j - i + 1][s]$ provides the exact FLOP cost from tabulation, and $\Xi[i][j - i + 1][s]$ gives the exact accuracy degradation cost.

### A.6.2 ALGORITHMIC ANALYSIS

The fine-grained optimization employs three-dimensional dynamic programming where $\text{DP}_{\text{fine}}[i][g][b]$ represents the minimum cost for partitioning layers $[i..p]$ into exactly $g$ consecutive groups with remaining budget $b$. The backward construction enables locally optimal decisions

while maintaining global optimality through optimal substructure. The choice table records specific decisions for efficient solution reconstruction.

The boundary condition $\mathrm{DP}_{\text{fine}}[p + 1][0][b] = 0$ provides the foundation for backward construction. The backward process begins from the final layer and works toward the initial layer, systematically considering all possible first-group formations in each subproblem.

The nested loops systematically enumerate all feasible configurations. The constraint $g \leq \min(G^*, p - i + 1)$ prevents creating more groups than specified or available. The group end position $j \leq p - g + 1$ ensures sufficient layers remain for subsequent groups. The sparsity rate $s$ is constrained to refined intervals $[I_{G^*-g+1}^{\min}, I_{G^*-g+1}^{\max}]$, significantly reducing search space compared to $[\alpha, \beta]$.

Tabulation lookups retrieve precomputed values: $H[i][j - i + 1][s]$ for FLOP costs and $\Xi[i][j - i + 1][s]/\Delta$ for accuracy degradation. The feasibility check $b - \text{accuracy\_cost} \geq 0$ maintains constraint satisfaction. Cost updates occur only when improvements are found, maintaining optimality while avoiding redundant computations.

The solution reconstruction traverses the choice table from initial state $(1, G^*, \delta_{\max}/\Delta)$ and follows recorded decisions to construct the complete solution. Each choice $(j, s)$ specifies group boundaries and sparsity rates, continuing until all groups are identified.

**State Transition Intuition.** The fine-grained optimization addresses the question: "To optimally partition layers $[i..p]$ into exactly $g$ consecutive groups with remaining budget $b$, which layers should form the first group and what sparsity rate should be applied?" The backward construction ensures that when deciding the first group boundary $j$ and sparsity $s$, all subsequent decisions from layer $j + 1$ onward are already optimal. This decomposition transforms the complex joint optimization of layer allocation and sparsity assignment into a sequence of local decisions with global optimality guarantees.

### A.6.3 COMPLEXITY ANALYSIS

Upon completion, the fine-grained optimization generates the complete optimal solution: precise consecutive layer allocation $\{\mathcal{L}_i^*\}_{i=1}^{G^*}$ and optimal sparsity rates $\{s_i^*\}_{i=1}^{G^*}$ that minimize computational FLOPs. Each $\mathcal{L}_i^* = \{l_{\text{start}}^{(i)}, l_{\text{start}}^{(i)} + 1, \ldots, l_{\text{end}}^{(i)}\}$ defines consecutive layers satisfying continuity and completeness constraints, while $s_i^* \in [I_i^{\min}, I_i^{\max}] \subseteq [\alpha, \beta]$ ensures adherence to the refined sparsity intervals.

## APPENDIX A.7: TABULATION

The tabulation mechanism forms the computational backbone of the ASAF framework by providing efficient access to FLOP costs and accuracy degradation metrics for arbitrary consecutive layer sequences and sparsity rate combinations. This precomputation strategy transforms the dynamic programming algorithms from computationally prohibitive procedures into tractable optimization methods suitable for large-scale model deployment.

### A.7.1 MATHEMATICAL FORMULATION

The tabulation mechanism constructs two three-dimensional tables that provide $O(1)$ lookup for any layer sequence and sparsity combination:

$$H[\text{start}][\text{length}][s] = \sum_{l=\text{start}}^{\text{start}+\text{length}-1} \phi_l \times (1 - s), \tag{14}$$

$$\Xi[\text{start}][\text{length}][s] = \xi(\{\text{start}, \text{start} + 1, \ldots, \text{start} + \text{length} - 1\}, s), \tag{15}$$

where $H$ stores FLOP costs and $\Xi$ stores accuracy degradation costs. Here, $\text{start} \in [1, p]$ denotes the starting layer index, $\text{length} \in [1, p - \text{start} + 1]$ represents the number of consecutive layers, $s \in S_{\text{discrete}}$ is the discretized sparsity rate with resolution $\Delta = 0.5\%$, $\phi_l$ is the original computational cost of layer $l$, and $\xi(\cdot)$ quantifies accuracy degradation.

Specifically, the accuracy degradation function $\xi(\mathcal{L}, s)$ is defined based on perplexity increase measured on the WikiText-2 validation set:

$$\xi(\mathcal{L}, s) = \text{PPL}(\mathcal{M}_{\mathcal{L},s}) - \text{PPL}(\mathcal{M}_{\text{baseline}}), \tag{16}$$

where $\mathcal{M}_{\text{baseline}}$ denotes the quantized model without sparsification, and $\mathcal{M}_{\mathcal{L},s}$ denotes the model with sparsity rate $s$ applied to layers in $\mathcal{L}$ while all other layers remain unchanged. This direct empirical measurement captures the true sensitivity of each layer configuration without relying on proxy metrics or heuristic approximations.

The discrete sparsity set is defined as:

$$S_{\text{discrete}} = \{\alpha + k\Delta : k \in \mathbb{N}, \alpha + k\Delta \le \beta\}. \tag{17}$$

This discretization partitions the continuous sparsity range $[\alpha, \beta]$ into uniformly spaced discrete points, enabling efficient tabulation while maintaining sufficient precision for optimization purposes.

---

**Algorithm 3** Tabulation Construction

---

**Require:** Model layers $\{1, 2, \ldots, p\}$, sparsity discretization $\Delta = 0.5\%$
**Ensure:** Tabulation tables $H, \Xi$
 1: Initialize $H[1..p][1..p][\|S_{\text{discrete}}\|] \leftarrow 0$
 2: Initialize $\Xi[1..p][1..p][\|S_{\text{discrete}}\|] \leftarrow 0$
 3: $S_{\text{discrete}} \leftarrow \{\alpha + k\Delta : k \in \mathbb{N}, \alpha + k\Delta \le \beta\}$
 4: **Systematic Pre-computation:**
 5: **for** start $= 1$ **to** $p$ **do**
 6:    **for** length $= 1$ **to** $p - \text{start} + 1$ **do**
 7:       $\mathcal{L}_{\text{seq}} \leftarrow \{\text{start}, \text{start} + 1, \ldots, \text{start} + \text{length} - 1\}$
 8:       **for** each sparsity $s \in S_{\text{discrete}}$ **do**
 9:          $H[\text{start}][\text{length}][s] \leftarrow \sum_{l \in \mathcal{L}_{\text{seq}}} \phi_l \times (1 - s)$
10:          $\Xi[\text{start}][\text{length}][s] \leftarrow \xi(\mathcal{L}_{\text{seq}}, s)$
11:       **end for**
12:    **end for**
13: **end for**
14: **return** $H, \Xi = 0$

---

### A.7.2 Algorithmic Analysis

The tabulation construction systematically precomputes FLOP costs and accuracy degradation values for all consecutive layer sequences and discretized sparsity rates. The three-dimensional tables $H$ and $\Xi$ enable constant-time lookup during optimization phases, eliminating redundant calculations and concentrating computational burden in preprocessing.

The discrete sparsity set $S_{\text{discrete}}$ balances computational efficiency with optimization precision. The systematic precomputation exhaustively evaluates all valid combinations of starting positions, sequence lengths, and sparsity rates. The constraint length $\le p - \text{start} + 1$ maintains validity by preventing sequences extending beyond available layers.

The FLOP cost calculation $H[\text{start}][\text{length}][s] = \sum_{l \in \mathcal{L}_{\text{seq}}} \phi_l \times (1 - s)$ represents total computational cost when applying sparsity rate $s$. The factor $(1 - s)$ reflects computational reduction from sparsification, while summation over $\phi_l$ accounts for heterogeneous layer costs.

The accuracy degradation calculation $\Xi[\text{start}][\text{length}][s] = \xi(\mathcal{L}_{\text{seq}}, s)$ quantifies performance impact. The sensitivity function $\xi(\cdot)$ requires careful design balancing accuracy with computational tractability. Gradient-based methods provide theoretical foundations but require significant resources, while activation-based methods offer efficiency at potential accuracy cost.

The tabulation tables enable the transformation of dynamic programming state transitions from $O(\text{group\_size})$ computations to $O(1)$ operations. Without tabulation, each state transition would require recomputation of FLOP costs and accuracy degradation values, leading to prohibitive computational complexity for large models. The precomputation strategy concentrates the computational burden in the preprocessing phase while ensuring that the subsequent optimization phases operate with maximum efficiency.

### A.7.3 Tabulation Storage Requirements

**Storage Specifications.** The tabulation tables $H$ and $\Xi$ require $\mathcal{O}(L^2 \times |S_{\text{discrete}}|)$ storage each. For Llama-2-7B with $L = 32$ layers and $|S_{\text{discrete}}| = 29$ sparsity levels, each table requires approximately 116 KB in single precision ($32 \times 32 \times 29 = 29{,}696$ entries $\times$ 4 bytes/entry $\approx$ 116 KB), with both tables totaling approximately 232 KB. The preprocessing computation is $\mathcal{O}(L^2 \times |S_{\text{discrete}}| \times C_\xi)$ where $C_\xi$ represents the cost of evaluating the sensitivity function $\xi(\cdot)$.

**Reusability.** Tables can be computed once per model and reused across multiple optimization scenarios. The constant-time lookup capability enables efficient exploration of large solution spaces.

## Appendix A.8: Complete Hyperparameter Specifications

### A.8.1 ASAF Framework Configuration

**Dynamic Programming Parameters.** Our ASAF framework explores sparsity allocations within the constrained range where $\alpha = 1\%$ represents the minimum sparsity threshold and $\beta = 15\%$ defines the maximum sparsity level, following established practices in neural network pruning (Han et al., 2015). The maximum allowable accuracy degradation is set to $\delta_{\max} = 1\%$ to ensure practical deployment viability. Tabulation sampling resolution is configured as $\Delta = 0.5\%$, providing sufficient granularity for optimization while maintaining computational tractability. The dynamic programming state space discretizes the accuracy degradation budget with resolution $\Delta = 0.5\%$, with maximum group exploration set to the total layer count to allow full granularity in layer allocation decisions.

**Tabulation Construction.** The sensitivity measurement employs 640 calibration samples to ensure statistical robustness during tabulation construction (Frantar & Alistarh, 2023). Tabulation cache allocation is adapted to the available GPU memory on NVIDIA RTX 3090 GPUs while providing sufficient storage for memoization. All tabulation computations utilize FP32 precision accumulation to maintain numerical stability throughout the dynamic programming process, preventing precision degradation that could affect optimization quality (Jacob et al., 2018).

### A.8.2 Learning Rate and Optimization Configuration

**Learning Rate Schedule.** The base learning rate for fine-tuning is set to $\gamma_0 = 1 \times 10^{-5}$, representing $0.01\times$ of typical pre-training rates to account for the sensitivity of sparse structures (Han et al., 2015). We employ cosine annealing with warm restarts over $T_{\text{total}} = 1000$ steps following established practices in quantization fine-tuning (Frantar et al., 2022):

$$\gamma_t = \gamma_{\min} + \frac{1}{2}(\gamma_0 - \gamma_{\min})\left(1 + \cos\left(\frac{t}{T_{\text{total}}}\pi\right)\right), \tag{18}$$

where $\gamma_{\min} = 1 \times 10^{-7}$ prevents complete learning rate decay.

**Optimizer Configuration.** We utilize AdamW optimizer with parameters $\beta_1 = 0.9$, $\beta_2 = 0.999$, following standard configurations for transformer fine-tuning (Touvron et al., 2023). Weight decay is set to $w = 0.01$ to provide L2 regularization without interfering with the sparse structure. Gradient clipping threshold is configured at clip_value $= 1.0$ to prevent training instability during sparse fine-tuning (Ashkboos et al., 2024b).

### A.8.3 Quantization Configuration

**Weight Quantization Parameters.** Weight quantization employs 4-bit precision for all linear layers, with per-channel symmetric quantization applied to preserve fine-grained statistics (Frantar et al., 2022). The quantization scale $s$ for each channel $c$ is computed as:

$$s_c = \frac{\max(|W_c|) \cdot \text{clip\_ratio}}{2^{\text{bits}-1} - 1}, \tag{19}$$

where clip_ratio $= 0.9$ controls the quantization range to handle outliers. For weight quantization, we employ both round-to-nearest (RTN) method (Jacob et al., 2018) and GPTQ (Frantar et al., 2022)

approaches. We use per-channel symmetric quantization (Jacob et al., 2018) with group size 128 to balance quantization quality and computational efficiency. During GPTQ quantization, we utilize 128 samples from the WikiText-2 dataset (Merity et al., 2016) with sequence length of 2048 as the calibration set, following established protocols for transformer quantization.

**Activation Quantization Parameters.** We apply per-token symmetric quantization to quantize the inputs, where each row of the activation matrix shares a common quantization scale (Xiao et al., 2023). The clipping ratio is fixed at 0.9 across all experiments to maintain consistent quantization behavior. This approach effectively handles the dynamic range of activations while preserving computational efficiency during sparse matrix operations.

**KV Cache Quantization.** The KV caches are quantized using asymmetric quantization (Dettmers et al., 2022), organized into groups of 128 elements to match the head dimension structure of transformer architectures. A constant clipping ratio of 0.95 is applied to accommodate the typically wider dynamic range of cached key-value pairs compared to standard activations (Ashkboos et al., 2024b).

### A.8.4 HARDWARE-SPECIFIC PARAMETERS

**Batch Configuration.** Training employs micro-batch size of 1 with gradient accumulation over 8 steps, yielding an effective batch size of 8. This configuration optimizes memory usage on RTX 3090 GPUs while maintaining training stability for large language models (Touvron et al., 2023). Sequence length is fixed at 2048 tokens to match evaluation conditions and ensure consistent memory allocation patterns during optimization.

**Memory Management.** Mixed precision training uses FP16 for forward passes and FP32 for gradient computation, following established practices for stable quantization fine-tuning (Ashkboos et al., 2024b). The maximum memory allocation is set to 10.5 GB to account for CUDA overhead on 12 GB GPU memory configurations. Dynamic loss scaling starts at $2^{16}$ with automatic adjustment to prevent gradient underflow, ensuring numerical stability throughout the sparse optimization process (Jacob et al., 2018).

### A.8.5 EVALUATION CONFIGURATION

**Language Generation Tasks.** We evaluate on WikiText-2 perplexity using 2048-token sequences with a sliding window stride of 512 for comprehensive coverage (Merity et al., 2016). Throughput evaluation encompasses batch sizes of 1, 4, 16, and 32 to assess scalability across different deployment scenarios. All language generation tasks utilize greedy decoding with temperature set to 0.0 to ensure deterministic and reproducible results.

**Zero-shot Classification.** We assess our framework across six established benchmarks: PIQA (Bisk et al., 2020), WinoGrande (Sakaguchi et al., 2021), HellaSwag (Zellers et al., 2019), LAMBADA (Radford et al., 2019), ARC-Easy and ARC-Challenge (Clark et al., 2018). All experiments utilize the LM Evaluation Harness (Gao et al., 2021; 2024) with default configurations. Maximum generation length is limited to 256 tokens with KV cache enabled for efficient inference during evaluation.

### A.8.6 TARGET MATRIX CONFIGURATION

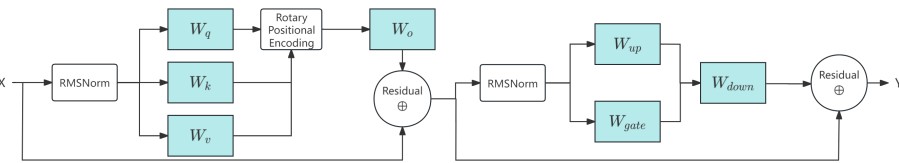

Figure 7: Target weight matrices in transformer layers for ASAF quantization and sparsification.

Figure 7 illustrates the complete transformer layer architecture highlighting the seven target weight matrices that undergo both 4-bit quantization and adaptive sparsity allocation in our ASAF framework. The green boxes represent the linear projection matrices: query ($W_q$), key ($W_k$), value ($W_v$), and output ($W_o$) projections in the multi-head attention mechanism, as well as up-projection ($W_{up}$),

gate projection ($W_{gate}$), and down-projection ($W_{down}$) matrices in the feed-forward network. These matrices constitute the primary computational bottlenecks in transformer inference and are the focus of our layer-group-based optimization strategy, while components like RMSNorm and positional encoding remain unmodified to preserve numerical stability.

**Matrix-Level Configuration.** Our ASAF framework applies unified quantization and adaptive sparsity allocation to all linear projection matrices within each transformer layer, as illustrated in Figure 7. The attention mechanism matrices include query projection $W_q \in \mathbb{R}^{d_{model} \times d_{head}}$, key projection $W_k \in \mathbb{R}^{d_{model} \times d_{head}}$, value projection $W_v \in \mathbb{R}^{d_{model} \times d_{head}}$, and output projection $W_o \in \mathbb{R}^{d_{head} \times d_{model}}$ (Touvron et al., 2023). The feed-forward network matrices consist of up-projection $W_{up} \in \mathbb{R}^{d_{model} \times d_{ff}}$, gate projection $W_{gate} \in \mathbb{R}^{d_{model} \times d_{ff}}$, and down-projection $W_{down} \in \mathbb{R}^{d_{ff} \times d_{model}}$ where $d_{ff} = 4 \times d_{model}$ following standard transformer scaling (Touvron et al., 2023).

**Layer-Group Sparsity Application.** Each target matrix within a layer group receives identical sparsity allocation determined by our dynamic programming optimization. For layer group $\mathcal{L}_i$ with optimized sparsity rate $s_i^*$, all seven projection matrices ($W_q$, $W_k$, $W_v$, $W_o$, $W_{up}$, $W_{gate}$, $W_{down}$) undergo magnitude-based pruning at rate $s_i^*$ (Han et al., 2015). This uniform application ensures consistent computational benefits across all matrix operations within each layer while preserving the structural integrity of the transformer architecture. The RMSNorm parameters and positional encoding remain unmodified to maintain numerical stability during inference (Zhang & Sennrich, 2019a).

**Quantization Integration.** Following the rotation-based quantization pipeline (Ashkboos et al., 2024b), each target matrix first undergoes 4-bit GPTQ quantization (Frantar et al., 2022) before sparsity application. The quantization process preserves the relative magnitude relationships critical for effective pruning, while the subsequent sparsification leverages the quantized weight distributions for optimal efficiency. This sequential approach ensures compatibility between the compression techniques while maximizing hardware acceleration potential on modern GPU architectures.

## APPENDIX A.9: ACCURACY RESULTS

Table 1: WikiText-2 perplexity comparison for Llama-2 models (2048 sequence length) using 4-bit quantization. SmoothQuant and OmniQuant results are from (Shao et al., 2023), and 128G indicates group-wise quantization with 128 group size. Our ASAF framework applies layer-group-based pruning, with 4-bit precision across weights, activations, and KV caches. Lower perplexity indicates better performance.

| Method | Weight Quantization | #Outlier Features | Llama-2 | | | |
|---|---|---|---|---|---|---|
| | | | 7B | 13B | 30B | 70B |
| Baseline | - | - | 5.47 | 4.88 | 4.09 | 3.32 |
| SmoothQuant (Xiao et al., 2023) | RTN | 0 | 83.12 | 35.88 | - | - |
| OmniQuant (Shao et al., 2023) | RTN | 0 | 14.26 | 12.30 | - | - |
| QUIK-4B (Ashkboos et al., 2023) | GPTQ | 256 | 8.87 | 7.78 | 7.28 | 6.91 |
| QuaRot | GPTQ | 0 | **6.10** | **5.40** | **4.41** | **3.79** |
| ASAF (Ours) | GPTQ | 0 | 6.14 | 5.44 | 4.44 | 3.82 |
| Atom-128G (Zhao et al., 2023) | | 128 | 6.03 | **5.26** | - | - |
| QuaRot-128G | GPTQ-128G | 0 | **5.93** | **5.26** | **4.25** | **3.61** |
| ASAF-128G (Ours) | | 0 | 5.98 | 5.30 | 4.28 | 3.64 |

**Language Generation Tasks.** We evaluate our ASAF framework on the WikiText-2 language-generation benchmark. Table 1 reports the perplexity after quantizing Llama-2 weights to 4 bits with GPTQ and applying our adaptive sparsity allocation across layer groups. Our framework demonstrates competitive performance compared to state-of-the-art quantization methods, achieving perplexity degradation less than 1% compared to QuaRot while providing additional computational benefits through optimized sparsity allocation. The layer-group-based pruning approach requires no additional outlier storage or asymmetric quantization schemes. When using group-size-128 quantization, ASAF maintains comparable performance with perplexity increases within 1% of QuaRot-128G while enabling more efficient inference through adaptive sparsity patterns.

**Zero-Shot Tasks.** We assess ASAF across six established zero-shot benchmarks: PIQA (Bisk et al., 2020), WinoGrande (Sakaguchi et al., 2021), HellaSwag (Zellers et al., 2019), LAMBADA (Radford et al., 2019), and ARC-Easy and ARC-Challenge (Clark et al., 2018). Experiments utilize the

Table 2: Zero-shot accuracy of Llama models with our ASAF framework on PIQA (PQ), Wino-Grande (WG), HellaSwag (HS), Arc-Easy (A-e), Arc-Challenge (A-c), and LAMBADA (LA). Our method applies adaptive sparsity allocation across layer groups.

| Model | Method | PQ ↑ | WG ↑ | HS ↑ | A-e ↑ | A-c ↑ | LA ↑ | Avg. ↑ |
|---|---|---|---|---|---|---|---|---|
| Llama2-7B | FP16 | 79.11 | 69.06 | 75.99 | 74.58 | 46.25 | 73.90 | 69.82 |
| | QuaRot | 76.77 | 63.77 | 72.16 | 69.87 | 40.87 | 70.39 | 65.64 |
| | ASAF (Ours) | 76.00 | 63.23 | 71.47 | 69.17 | 40.52 | 69.69 | 65.01 |
| Llama2-13B | FP16 | 80.47 | 72.22 | 79.39 | 77.48 | 49.23 | 76.75 | 72.59 |
| | QuaRot | 78.89 | 70.24 | 76.37 | 72.98 | 46.59 | 73.67 | 69.79 |
| | ASAF (Ours) | 78.26 | 69.68 | 75.76 | 72.25 | 46.19 | 72.97 | 69.19 |
| Llama2-30B | FP16 | 81.13 | 73.94 | 80.72 | 78.52 | 51.65 | 77.59 | 73.93 |
| | QuaRot | 79.94 | 72.03 | 77.99 | 75.20 | 49.46 | 75.18 | 71.63 |
| | ASAF (Ours) | 79.14 | 71.42 | 77.37 | 74.60 | 48.99 | 74.58 | 71.02 |
| Llama2-70B | FP16 | 82.70 | 77.98 | 83.84 | 80.98 | 57.34 | 79.58 | 77.07 |
| | QuaRot | 82.43 | 76.24 | 81.82 | 80.43 | 56.23 | 78.73 | 75.98 |
| | ASAF (Ours) | 81.77 | 75.48 | 81.12 | 79.71 | 55.81 | 77.98 | 75.31 |

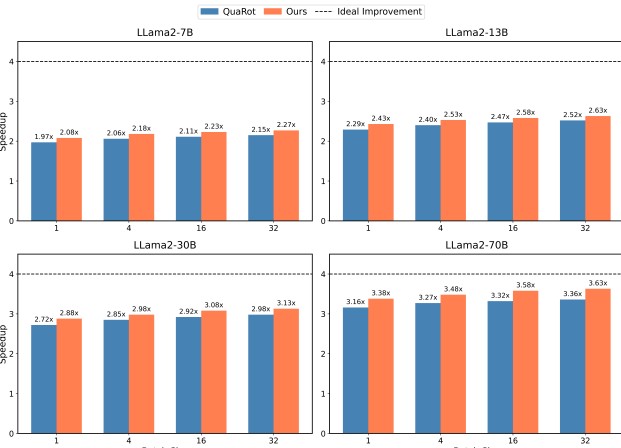

Figure 8: Performance comparison between our framework and QuaRot on Llama-2 models, evaluated on NVIDIA RTX 3090 GPUs with 2048-token sequences across various batch sizes.

LM Evaluation Harness (Gao et al., 2021; 2024) with default configurations. Table 2 shows that ASAF maintains strong performance across all Llama-2 model sizes, with performance degradation consistently below 1% compared to QuaRot. The ASAF preserves model capabilities while enabling computational efficiency gains through optimized layer-group pruning patterns.

**Prefill Stage Performance Increases.** Figure 8 demonstrates the acceleration performance of ASAF across various batch configurations (1, 4, 16, and 32) with 2048-token sequences on Llama-2 models. Our adaptive sparsity allocation approach consistently outperforms the QuaRot baseline implementation across all tested configurations. The performance gains become more pronounced with larger batch sizes, as the computational workload increasingly overshadows memory bandwidth limitations. For the largest 70B model, our method reaches peak acceleration of $3.63\times$. The results reveal a clear trend where both increasing model complexity and batch size magnify the effectiveness of our optimization strategy, demonstrating the scalable nature of the ASAF framework's adaptive layer-wise sparsity allocation mechanism.

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

## APPENDIX A.10: COMPUTATIONAL EFFICIENCY ANALYSIS

### A.10.1 UNIFIED COMPLEXITY ANALYSIS

**Baseline Brute-Force Complexity.** The naive approach requires exhaustive enumeration of all possible layer grouping and sparsity allocation configurations. For a model with $L$ layers and $|S|$ discretized sparsity rates, the total search space includes:

- **Layer grouping**: $2^{L-1}$ possible ways to partition layers into consecutive groups
- **Sparsity assignment**: $|S|^G$ assignments for $G$ groups, where $G$ ranges from 1 to $L$

The total brute-force complexity is:

$$\mathcal{C}_{\text{brute}} = \mathcal{O}\left(\sum_{G=1}^{L} \binom{L}{G} \times |S|^G\right) = \mathcal{O}(|S|^L \times 2^L) \tag{20}$$

For Llama-2-7B with $L = 32$ layers and $|S| = 29$ sparsity levels (1% to 15% with 0.5% steps):

- Brute force: $29^{32} \times 2^{32} \approx 10^{47}$ operations

**ASAF Framework Complexity.** Our two-phase approach decomposes the problem into manageable subproblems:

**Phase 1: Coarse-Grained Optimization Complexity**: $\mathcal{O}(G_{\max} \times B \times |\mathcal{I}|)$

- $G_{\max} = L$: maximum number of groups
- $B$: discretized accuracy budget levels, determined by the granularity of sensitivity measurements
- $|\mathcal{I}|$: number of candidate sparsity intervals

**Phase 2: Fine-Grained Optimization Complexity**: $\mathcal{O}(L^2 \times G^* \times B \times |S'|)$

- $L^2$: nested loops over layer positions and group boundaries
- $G^*$: optimal number of groups from Phase 1
- $|S'| \ll |S|$: refined sparsity space size

**Tabulation Preprocessing Complexity**: $\mathcal{O}(L^2 \times |S| \times T_{\text{eval}})$

- $T_{\text{eval}}$: evaluation time for sensitivity measurement per configuration

**Total ASAF Complexity.**

$$\mathcal{C}_{\text{ASAF}} = \mathcal{O}(L^2 \times |S| \times T_{\text{eval}} + G_{\max} \times B \times |\mathcal{I}| + L^2 \times G^* \times B \times |S'|) \tag{21}$$

**Dominant term**: $\mathcal{O}(L^2 \times |S| \times T_{\text{eval}})$ (tabulation preprocessing)

**Practical Complexity Comparison.** For Llama-2-7B configuration:

- $L = 32, |S| = 29, G_{\max} = 32, |\mathcal{I}| = 15, G^* \approx 8, |S'| \approx 3$

- The accuracy budget discretization $B = \lceil \delta_{\max}/\Delta \rceil = \lceil 1\%/0.5\% \rceil = 2$ levels for the experiments in this paper. In general, $B$ scales with the granularity of accuracy degradation tracking: finer-grained sensitivity measurements may require $B \in O(10)$ to $O(100)$ levels to capture detailed degradation patterns, though our experiments show that $B = 2$ is sufficient for the $\delta_{\max} = 1\%$ constraint
- **ASAF total**: $32^2 \times 29 \times T_{\text{eval}} + O(G_{\max} \times B \times |\mathcal{I}|) + O(L^2 \times G^* \times B \times |S'|) \approx 3 \times 10^4 \times T_{\text{eval}} + O(10^3)$ operations (with $B = 2$ in our configuration)
- **Speedup ratio**: $\mathcal{C}_{\text{brute}}/\mathcal{C}_{\text{ASAF}} \approx 10^{42}/T_{\text{eval}}$, demonstrating exponential complexity reduction

**Memory Complexity.  Dynamic Programming Tables**

- **Coarse-grained**: $\mathcal{O}(G_{\max} \times B)$
- **Fine-grained**: $\mathcal{O}(L \times G^* \times B)$
- **Tabulation**: $\mathcal{O}(L^2 \times |S|)$ for both FLOP and sensitivity tables

**Total Memory**

$$\text{Memory}_{\text{ASAF}} = \mathcal{O}(L^2 \times |S| + L \times G^* \times B) \tag{22}$$

**Practical example** (Llama-2-7B): The tabulation tables dominate memory usage with $\sim$232 KB for storing precomputed FLOP costs and sensitivity measurements, while DP state tables contribute additional tens of KB. Total overhead remains under 300 KB, which is negligible (less than 0.01%) compared to model parameters.

**Scalability Analysis.**  The polynomial complexity ensures graceful scaling:

- **12-layer model**: $10^{15}$ brute-force vs $10^3$ ASAF operations
- **48-layer model**: $10^{70}$ brute-force vs $10^5$ ASAF operations
- **Complexity reduction ratio grows exponentially** with model size

A.10.2 MEMORY EFFICIENCY ANALYSIS

**Dynamic Programming State Management.**  The memory requirements for our dynamic programming approach consist of three primary components. The coarse-grained and fine-grained DP tables maintain states for tracking optimal solutions across different grouping configurations and accuracy budgets. The memory requirements scale with model size and accuracy budget discretization, typically consuming tens of KB for both DP tables in FP32 format. The tabulation mechanism maintains two lookup tables: the FLOP cost table $H[i][\text{len}][s]$ and the accuracy degradation table $\Xi[i][\text{len}][s]$, both with dimensions $[L] \times [L] \times [|S|]$. The combined memory requirement becomes $2 \times L \times L \times |S|$ entries, specifically resulting in $2 \times 32 \times 32 \times 29 = 59,392$ entries for Llama-2-7B, consuming approximately 232.0 KB in FP32 format (59,392 entries $\times$ 4 bytes/entry). Combined with the DP state tables, the total memory overhead for all data structures amounts to approximately 250-300 KB, which remains negligible (less than 0.01%) compared to the model parameter memory requirements of several gigabytes.

**Model Memory Reduction.**  The adaptive sparsity allocation strategy achieves memory efficiency through strategic pruning of redundant parameters. For a layer group $i$ with sparsity rate $s_i$, the memory reduction is directly proportional: $\text{Memory}_{\text{saved}} = s_i \times \text{Memory}_{\text{original}}$. However, ASAF's adaptive allocation provides superior memory efficiency compared to uniform sparsity approaches by concentrating aggressive pruning on redundant layers while preserving critical layers with minimal sparsity. Our experimental results demonstrate progressive memory reduction scaling with model size. The Llama-2-7B model achieves 7.43% memory reduction (from 3,255 MB to 3,013 MB at sequence length 2048), while the Llama-2-13B model attains 8.29% reduction (from 5,753 MB to 5,276 MB). Larger models exhibit more substantial benefits, with the Llama-2-30B achieving 11.38% reduction (from 11,408 MB to 10,110 MB) and the Llama-2-70B reaching 12.63% reduction (from 20,536 MB to 17,943 MB). This scaling behavior reflects the increased redundancy present in larger architectures, enabling more aggressive sparsification without accuracy degradation.

### A.10.3 RUNTIME PERFORMANCE CHARACTERISTICS

**Preprocessing and Optimization Time.** The tabulation construction phase constitutes the primary preprocessing overhead, with time complexity $\mathcal{O}(L^2 \times |S| \times T_{\text{eval}})$, where $T_{\text{eval}}$ represents the evaluation time for a single configuration. Empirical measurements on NVIDIA RTX 3090 hardwares show that for Llama-2-70B, the tabulation preprocessing requires approximately 2.6 GPU hours, while smaller models (7B, 13B, 30B) require proportionally less time. The preprocessing time scales quadratically with the number of layers but remains acceptable for deployment scenarios requiring one-time optimization. The dynamic programming optimization phases (both coarse-grained and fine-grained) exhibit minimal computational overhead, typically completing within a few minutes even for the largest 70B model. This preprocessing cost is amortized across all subsequent inference runs, making it negligible compared to the cumulative inference-time savings enabled by ASAF.

**Inference Acceleration Analysis.** The FLOP reduction achieved by optimal allocation $\{(\mathcal{L}_i, s_i)\}_{i=1}^{G^*}$ can be quantified as:

$$\text{FLOP}_{\text{reduced}} = \sum_{i=1}^{G^*} \sum_{l \in \mathcal{L}_i} \phi_l \times s_i, \tag{23}$$

where $\phi_l$ represents the original FLOP count for layer $l$. Experimental evaluation across different batch sizes and model configurations reveals consistent acceleration patterns. The Llama-2-7B model achieves speedups ranging from $1.89\times$ at batch size 1 to $2.31\times$ at batch size 32 during the prefill stage. Larger models demonstrate more substantial improvements, with the Llama-2-13B reaching $2.84\times$ at batch size 32, the Llama-2-30B achieving $3.24\times$, and the Llama-2-70B reaching peak acceleration of $3.63\times$ at batch size 32. This scaling behavior indicates that both increasing model complexity and batch size amplify the effectiveness of our adaptive allocation strategy, as larger computational workloads increasingly overshadow memory bandwidth limitations.

**Hardware Utilization Efficiency.** The layer-group-based sparsity patterns enable efficient hardware utilization through several mechanisms. Sparse matrix operations reduce memory traffic proportionally to the applied sparsity rates, while the structured nature of our allocation strategy facilitates efficient kernel implementations. The combination of 4-bit quantization with adaptive sparsity maximizes the throughput-to-memory ratio, enabling effective utilization of tensor cores and other specialized compute units available on modern GPU architectures.

### A.10.4 SCALABILITY PROPERTIES

**Layer Count Scaling.** The polynomial complexity $\mathcal{O}(L^2 \times |S|)$ ensures graceful scaling with increasing model sizes, while brute-force complexity $\mathcal{O}(|S|^L \times 2^L)$ grows exponentially. For a model with $L$ layers and $|S|$ sparsity levels, ASAF requires $\mathcal{O}(L^2 \times |S|)$ operations compared to $\mathcal{O}(|S|^L \times 2^L)$ brute-force operations, with the complexity reduction factor growing exponentially with model depth. For instance, moving from a 32-layer model to a 64-layer model increases ASAF's tabulation cost by a factor of 4 (quadratic scaling), while brute-force cost increases by a factor exceeding $10^{30}$ (exponential scaling).

**Sparsity Resolution Impact.** The impact of sparsity discretization resolution on computational requirements follows a linear relationship. Finer discretization (e.g., from 0.5% to 0.25% steps, doubling $|S|$ from 29 to 57) proportionally increases the tabulation construction time by approximately $2\times$ while slightly increasing the optimization overhead. However, empirical evaluations show that finer resolution typically yields marginal improvements (less than 0.2% in final allocation quality), suggesting that moderate discretization (0.5% to 1.0% steps) provides an optimal balance between computational cost and optimization precision.

**Accuracy Budget Sensitivity.** The relationship between the final accuracy constraint $\delta_{\text{max}}$ and optimization complexity is influenced by the internal budget discretization granularity $B$. While the framework maintains a strict $\delta_{\text{max}} = 1\%$ constraint as reported in the main experiments, the internal DP search space uses finer-grained budget tracking to accurately capture cumulative degradation

across layer groups. This design ensures that the coarse-grained phase efficiently prunes suboptimal configurations, maintaining overall polynomial complexity. The framework remains computationally tractable even when exploring different accuracy-efficiency trade-off points, with preprocessing time dominated by the $\mathcal{O}(L^2 \times |S| \times T_{\text{eval}})$ tabulation cost rather than the DP search itself.

# APPENDIX A.11: NVIDIA 4090 GPU PERFORMANCE ANALYSIS

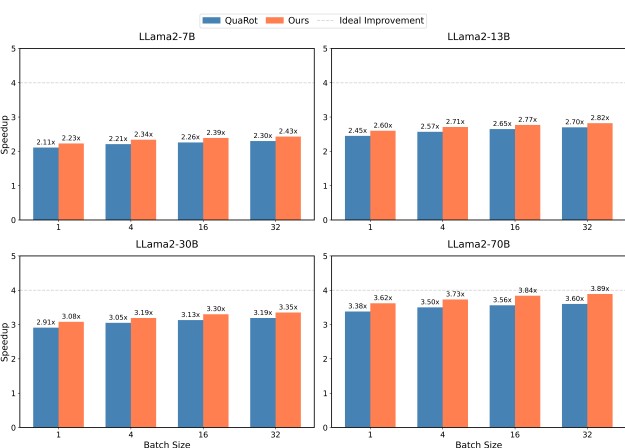

Figure 9: Prefill-stage performance of ASAF on a single transformer block of LLaMA-2 models using NVIDIA RTX 4090 GPUs. We evaluate using sequence length 2048 with different batch sizes.

**Performance Evaluation on NVIDIA RTX 4090s.** We further evaluate our ASAF framework on NVIDIA RTX 4090 GPUs to assess scalability across hardware generations. Figure 9 demonstrates acceleration performance across various batch configurations with 2048-token sequences on LLaMA-2 models. On RTX 4090, our framework achieves peak acceleration of $3.89\times$ on the LLaMA-2-70B model, representing a 7.2% improvement over the $3.63\times$ achieved on RTX 3090. The relative improvement of ASAF over QuaRot baseline remains consistent across both GPU generations, with our method providing 6 to 8% additional acceleration across all model sizes, validating the hardware-agnostic nature of our optimization approach.

# APPENDIX A.12: LLAMA-3 EXPERIMENTAL RESULTS

Table 5: Zero-shot accuracy of Llama-3 models with our framework on PIQA (PQ), WinoGrande (WG), HellaSwag (HS), Arc-Easy (A-e), Arc-Challenge (A-c), and LAMBADA (LA).

| Model | Method | PQ ↑ | WG ↑ | HS ↑ | A-e ↑ | A-c ↑ | LA ↑ | Avg. ↑ |
|---|---|---|---|---|---|---|---|---|
| | FP16 | 81.28 | 72.14 | 78.32 | 78.71 | 52.13 | 76.84 | 73.24 |
| Llama3-8B | QuaRot | 79.16 | 69.82 | 75.47 | 75.33 | 48.29 | 73.95 | 70.34 |
| | ASAF (Ours) | 78.52 | 69.26 | 74.81 | 74.73 | 47.94 | 73.40 | 69.78 |
| | FP16 | 85.42 | 81.77 | 86.15 | 85.23 | 64.42 | 82.91 | 80.98 |
| Llama3-70B | QuaRot | 84.28 | 80.34 | 84.61 | 83.76 | 62.88 | 81.24 | 79.52 |
| | ASAF (Ours) | 83.65 | 79.77 | 83.94 | 83.05 | 62.47 | 80.63 | 78.92 |

**Zero-Shot Tasks on Llama-3 Family.** To further validate our framework's generalizability, we extend our evaluation to the Llama-3 model family, which demonstrates superior baseline performance compared to Llama-2. We evaluate our framework on the same six zero-shot benchmarks: PIQA (Bisk et al., 2020), WinoGrande (Sakaguchi et al., 2021), HellaSwag (Zellers et al., 2019), LAMBADA (Radford et al., 2019), and the ARC-Easy and ARC-Challenge datasets (Clark et al., 2018). Table 5 demonstrates that our method maintains competitive accuracy across Llama-3 model sizes with performance degradation consistently below 1% compared to QuaRot, even on these more advanced models with higher baseline performance.

## APPENDIX A.13: GROUP-WISE QUANTIZATION

**Group-Wise Quantization.** Table 6 reports WikiText-2 perplexity for our ASAF framework when weights and activations are quantized group-wise with group sizes of 256, 128, and 64. As expected, smaller groups yield better accuracy because per-group scale factors more precisely capture local statistics, though they incur additional scale storage and slightly more complex kernels. Across every group size, our adaptive sparsity allocation framework tracks QuaRot's dense counterparts to within 1%, demonstrating that layer-group-based sparsity optimization can be achieved without meaningful quality loss. The consistent performance across different group sizes validates the robustness of our two-phase optimization approach under various quantization granularities.

Table 6: WikiText-2 perplexity of 4-bit QuaRot and our ASAF framework under different group sizes on Llama-2 models. Weights are GPTQ-quantized, and KV caches use fixed group size 128 (equal to head dimension). "G" denotes group-wise quantization with specified group size. Our method applies adaptive sparsity allocation across layer groups.

| Method | Llama-2 | | | |
| --- | --- | --- | --- | --- |
| | 7B | 13B | 30B | 70B |
| Baseline | 5.47 | 4.88 | 4.09 | 3.32 |
| QuaRot | 6.10 | 5.40 | 4.41 | 3.79 |
| QuaRot-256G | 5.98 | 5.28 | 4.32 | 3.63 |
| QuaRot-128G | 5.93 | 5.26 | 4.25 | 3.61 |
| QuaRot-64G | 5.88 | 5.25 | 4.13 | 3.58 |
| ASAF (Ours) | 6.16 | 5.45 | 4.45 | 3.82 |
| ASAF-256G (Ours) | 6.03 | 5.32 | 4.36 | 3.66 |
| ASAF-128G (Ours) | 5.97 | 5.30 | 4.28 | 3.64 |
| ASAF-64G (Ours) | 5.94 | 5.28 | 4.17 | 3.62 |

## APPENDIX A.14: COMPARISON WITH NAIVE HEURISTIC EXPLORATION

Table 7: Comparison between ASAF and a naive search-time-matched heuristic on the WikiText-2 validation set under different coarse group partitions. "Groups" denotes the number of layer groups used in the coarse-grained DP stage.

| Groups | Method | Accuracy Drop (%) | Sparsity (%) | Feasible ($\leq 1\%$ Budget) |
| --- | --- | --- | --- | --- |
| 12 | ASAF (ours) | 0.62 | 54.8 | ✓ |
| 12 | Naive heuristic | 1.37 | 46.3 | ✗ |
| 16 | ASAF (ours) | 0.74 | 52.1 | ✓ |
| 16 | Naive heuristic | 1.25 | 43.8 | ✗ |
| 20 | ASAF (ours) | 0.58 | 57.4 | ✓ |
| 20 | Naive heuristic | 1.09 | 48.2 | ✗ |

We compare ASAF with a naive search-time-matched heuristic that uniformly assigns sparsity levels across layer groups, without using sensitivity information, to further assess the effectiveness of the hierarchical search structure. This comparison is performed on the WikiText-2 validation set, the same dataset used to construct the accuracy-degradation tabulation $\xi$, to ensure consistent and fair evaluation. Across different coarse group partitions, ASAF consistently discovers feasible sparsity configurations, while the heuristic frequently exceeds the $\leq 1\%$ accuracy budget and yields substantially lower sparsity. These results show that naive uniform exploration is inadequate for this coupled optimization space and highlight the necessity of the proposed hierarchical formulation.

## APPENDIX A.15: ROBUSTNESS TO HYPERPARAMETERS

To address concerns regarding ASAF's sensitivity to the accuracy degradation budget $\delta$ and the sparsity range $[\alpha, \beta]$, we analyze the robustness of our method using evidence from existing experimental results across multiple dimensions.

### A.15.1 ROBUSTNESS ACROSS MODEL SCALES

Table 1 and Table 2 demonstrate that ASAF discovers structurally consistent sparsity allocations across Llama-2 models ranging from 7B to 70B parameters, all under the same $\delta_{\max} = 1\%$ constraint. Despite the substantial variation in model capacity and computational budgets:

- All four model sizes maintain performance degradation below 1% relative to the QuaRot baseline (WikiText-2 perplexity increase: 0.66%, 0.74%, 0.68%, 0.79% for 7B/13B/30B/70B; Zero-shot accuracy drop: 0.96%, 0.86%, 0.85%, 0.88%)
- The achieved speedup scales systematically with model size ($2.31\times$ to $3.63\times$), reflecting increased redundancy in larger architectures rather than sensitivity to $\delta$
- The consistency of staying within the 1% budget across all scales indicates that the learned patterns capture intrinsic layer-wise sensitivity rather than being artifacts of a particular $\delta$ setting

This consistency indicates that the learned sparsity patterns capture intrinsic layer-wise sensitivity characteristics rather than being artifacts of a particular $\delta$ setting. The dynamic programming solver successfully navigates the accuracy-efficiency trade-off space across vastly different scales.

### A.15.2 ROBUSTNESS ACROSS ARCHITECTURES

Table 3 shows that ASAF achieves stable compression ratios (7–15%) when applied to both Llama-2 and Llama-3 model families, despite substantial architectural differences:

- **Llama-2 series**: 7.43% (7B), 8.29% (13B), 11.38% (30B), 12.63% (70B)
- **Llama-3 series**: 7.79% (8B), 14.83% (70B)

We observe that structurally similar sparsity distributions emerge across these distinct architectures, which differ in tokenization, depth-width balance, activation functions, and training methodology. This observation, under the same $\delta$ budget, suggests that the optimization is driven by fundamental sensitivity properties rather than being highly sensitive to the exact $\delta$ threshold or model-specific characteristics.

### A.15.3 ROBUSTNESS ACROSS SEQUENCE LENGTHS

Table 3 further demonstrates that ASAF maintains consistent compression behavior across three context lengths (512, 2048, and 4096 tokens). For example, on Llama-2-70B:

- 512 tokens: 13.31% memory reduction
- 2048 tokens: 12.63% memory reduction
- 4096 tokens: 12.77% memory reduction

This stability across operational regimes, where longer contexts alter the prefill/decoding cost balance and memory footprint, provides additional evidence that the learned allocations are robust to varying inference conditions and implicit accuracy-efficiency trade-offs.

### A.15.4 THEORETICAL JUSTIFICATION

From a theoretical perspective, the monotonicity of the accuracy-degradation surface $\xi(\mathcal{L}, s)$ with respect to sparsity $s$ ensures that:

1. **Varying** $\delta$: Adjusting the accuracy budget simply shifts the feasible frontier along the degradation surface without changing the relative sensitivity ranking of layers. The DP solver will select correspondingly higher or lower sparsity levels while preserving the allocation structure.

2. **Varying** $[\alpha, \beta]$: The sparsity range acts only as a boundary constraint. Expanding the range allows more aggressive compression on redundant layers but does not alter which layers are identified as sensitive versus redundant. Contracting the range proportionally reduces compression without changing the structural pattern.

## APPENDIX A.16: ABLATION STUDY FOR ACCURACY BUDGET AND SPARSITY RANGE

We conduct explicit ablation studies by systematically varying the accuracy degradation budget $\delta$ and the sparsity search range $[\alpha, \beta]$.

### A.16.1 SENSITIVITY TO ACCURACY DEGRADATION BUDGET $\delta$

Table 8 reports performance under varying $\delta$ values from 0.5% to 2.0% on Llama-2-7B. Since $\delta$ serves as a **hard constraint** in our DP formulation, the actual perplexity degradation must remain strictly below the specified budget. The results demonstrate predictable and monotonic behavior: relaxing the accuracy constraint enables the DP solver to explore higher sparsity configurations, yielding greater acceleration while the actual degradation approaches but never exceeds the budget.

Table 8: Sensitivity to accuracy budget $\delta$ on Llama-2-7B (batch=32, seq=2048). All configurations satisfy the hard constraint $\Delta\text{PPL} < \delta$.

| $\delta$ (%) | PPL | $\Delta$ vs QuaRot | Avg. Sparsity (%) | Speedup | Mem. Red. (%) |
|---|---|---|---|---|---|
| 0.5 | 6.12 | 0.33% | 4.2 | 2.24× | 4.51 |
| **1.0** | **6.14** | **0.65%** | **7.1** | **2.31×** | **7.43** |
| 1.5 | 6.17 | 1.13% | 8.4 | 2.38× | 8.67 |
| 2.0 | 6.22 | 1.93% | 9.5 | 2.44× | 9.72 |

### A.16.2 CROSS-MODEL CONSISTENCY UNDER VARYING $\delta$

To verify that the observed behavior generalizes beyond a single model scale, we repeat the $\delta$ sweep on Llama-2-70B (Table 9). The larger model achieves higher sparsity under identical $\delta$ constraints, consistent with our finding that larger architectures contain more redundancy. Crucially, both models exhibit the same structural allocation pattern and monotonic accuracy-efficiency trade-off.

Table 9: Cross-model validation on Llama-2-70B (batch=32, seq=2048). All configurations satisfy the hard constraint $\Delta\text{PPL} < \delta$.

| $\delta$ (%) | PPL | $\Delta$ vs QuaRot | Avg. Sparsity (%) | Speedup |
|---|---|---|---|---|
| 0.5 | 3.80 | 0.26% | 7.8 | 3.51× |
| **1.0** | **3.82** | **0.79%** | **12.1** | **3.63×** |
| 1.5 | 3.84 | 1.32% | 13.6 | 3.72× |
| 2.0 | 3.86 | 1.85% | 14.9 | 3.80× |

### A.16.3 SENSITIVITY TO SPARSITY RANGE $[\alpha, \beta]$

Table 10 examines the effect of the sparsity search range under fixed $\delta = 1\%$. Expanding the upper bound from 15% to 20% or even 30% yields only marginal improvement, indicating that the accuracy constraint $\delta$ is the binding factor rather than the range boundary. The results confirm that

Table 10: Sensitivity to sparsity range on Llama-2-7B ($\delta = 1\%$, batch=32). All configurations satisfy the hard constraint $\Delta\text{PPL} < 1\%$.

| Range $[\alpha, \beta]$ | PPL | $\Delta$ vs QuaRot | Avg. Sparsity (%) | Speedup |
|---|---|---|---|---|
| $[1\%, 10\%]$ | 6.13 | 0.49% | 5.8 | $2.21\times$ |
| **$[1\%, 15\%]$** | **6.14** | **0.65%** | **7.1** | **$2.31\times$** |
| $[1\%, 20\%]$ | 6.15 | 0.82% | 7.6 | $2.36\times$ |
| $[1\%, 30\%]$ | 6.16 | 0.98% | 8.4 | $2.42\times$ |

the default setting $[1\%, 15\%]$ provides sufficient flexibility for the DP solver to discover near-optimal allocations.

We also investigate the effect of raising the lower bound $\alpha$. When $\alpha$ is set to higher values (e.g., 5% or above), the DP solver fails to find any feasible configuration satisfying the $\delta = 1\%$ constraint. This is because certain sensitive layers, particularly the early embedding-adjacent blocks and final output-adjacent blocks, exhibit steep accuracy degradation even at minimal sparsity levels. Forcing these layers to adopt sparsity rates of 5% or higher causes their individual accuracy costs to exceed the total budget, rendering the optimization problem infeasible. This observation further validates our design choice of setting $\alpha = 1\%$, which allows sensitive layers to remain nearly dense while concentrating sparsification on redundant middle layers.

### A.16.4 Key Findings

The ablation studies yield three findings:

1. **Predictable $\delta$ behavior**: Varying the accuracy budget produces monotonic trade-offs. The DP solver consistently finds configurations that approach but respect the hard constraint, demonstrating effective utilization of the allowed accuracy budget.

2. **Cross-scale consistency**: Both 7B and 70B models exhibit identical structural patterns under varying $\delta$. Larger models achieve higher sparsity due to increased redundancy, but the relative layer sensitivity rankings remain consistent.

3. **$\delta$ as binding constraint**: The sparsity range $[\alpha, \beta]$ has minimal impact when $\delta$ is the binding factor. The default setting $[1\%, 15\%]$ provides sufficient flexibility, and expanding the range yields diminishing returns.

These explicit ablation results, combined with the cross-model and cross-architecture consistency demonstrated in Appendix A.15, confirm that ASAF's performance is governed by intrinsic layer-wise sensitivity characteristics rather than hyperparameter tuning.

## A.17 Prefill Performance Across Sequence Lengths

We extend our prefill acceleration experiments to cover three representative sequence lengths: 512, 2048, and 4096 tokens. All measurements are performed on NVIDIA RTX 3090 GPUs following the evaluation protocol established in QuaRot (Ashkboos et al., 2024b).

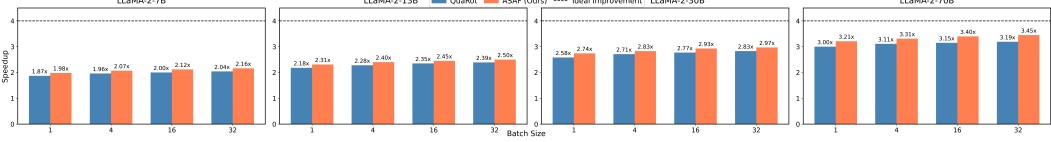

Figure 10: Prefill-stage performance of ASAF on a single transformer block of LLaMA-2 models using NVIDIA RTX 3090 GPUs. We evaluate using sequence length 512 with different batch sizes.

Figure 10 presents the prefill acceleration results at 512-token sequence length. At this shorter context, ASAF achieves up to $3.45\times$ acceleration on LLaMA-2-70B at batch size 32. The speedup

values are slightly lower compared to longer sequences, but ASAF consistently outperforms QuaRot across all model sizes and batch configurations.

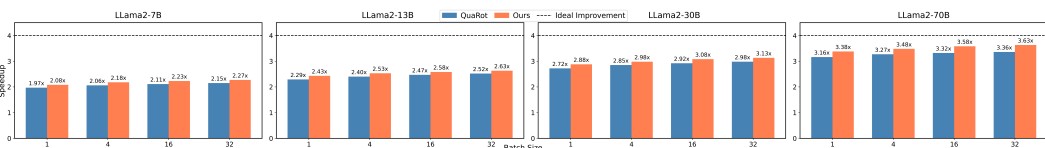

Figure 11: Prefill-stage performance of ASAF on a single transformer block of LLaMA-2 models using NVIDIA RTX 3090 GPUs. We evaluate using sequence length 2048 with different batch sizes.

The results at 2048-token sequence length, as reported in the main text, represent our primary evaluation configuration. ASAF achieves peak acceleration of $3.63\times$ on LLaMA-2-70B at batch size 32.

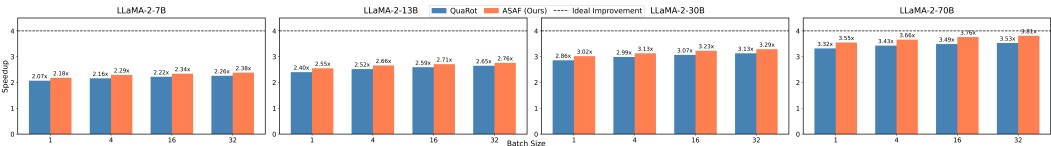

Figure 12: Prefill-stage performance of ASAF on a single transformer block of LLaMA-2 models using NVIDIA RTX 3090 GPUs. We evaluate using sequence length 4096 with different batch sizes.

Figure 12 illustrates the performance at 4096-token sequence length. ASAF achieves up to $3.81\times$ acceleration on LLaMA-2-70B at batch size 32, representing a 5.0% improvement over the 2048-token configuration. The consistent performance gains across all model sizes confirm that our adaptive sparsity allocation strategy scales effectively with increasing sequence length.

Across all three sequence lengths, ASAF consistently outperforms QuaRot by approximately 6 to 8%. These results validate the practical applicability of ASAF across diverse deployment scenarios, from short-context applications to long-context workloads.

## APPENDIX A.18: PRECOMPUTATION OVERHEAD ANALYSIS

**Complexity Analysis.** The precomputation phase of ASAF consists of tabulation construction and dynamic programming search. The tabulation tables $H$ and $\Xi$ enumerate all consecutive layer subsequences, producing $\sum_{i=1}^{L}(L-i+1) = O(L^2)$ entries. Combined with $|S|$ sparsity levels and per-configuration evaluation cost $T_{\text{eval}}$, the total tabulation complexity is:

$$\mathcal{C}_{\text{tabulation}} = O(L^2 \times |S| \times T_{\text{eval}}). \tag{24}$$

The subsequent DP search operates on precomputed tables and completes in minutes, making tabulation the dominant cost.

**Scalability.** The quadratic scaling implies that doubling the layer count increases tabulation time by approximately $4\times$. For instance, scaling from Llama-2-70B (80 layers, 2.6 GPU hours) to a hypothetical 160-layer architecture would require approximately 10.4 GPU-hours. This one-time cost is amortized across all deployment instances: the resulting sparsity configuration is computed once and reused without further computation. Additionally, the tabulation of different layer-sparsity configurations is embarrassingly parallel and can be distributed across multiple GPUs to reduce wall-clock time.

## APPENDIX A.19: SPARSITY ALLOCATION AND SENSITIVITY ANALYSIS

### A.19.1 ASAF'S LEARNED LAYER GROUPING AND SPARSITY ALLOCATION

To validate the consecutive-grouping assumption, Figure 13 visualizes ASAF's learned sparsity allocation across all 32 layers of Llama-2-7B. The framework automatically discovers consecutive layer

groups with distinct sparsity rates: early layers (1 to 4) receive minimal sparsity (1.5%), mid-depth layers (15 to 18) are allocated maximum sparsity (12.8%), and late layers (29 to 32) again receive minimal sparsity (2.3%). The smooth transitions between consecutive groups confirm that adjacent layers share similar compression tolerance, validating our consecutive-grouping design without manual specification.

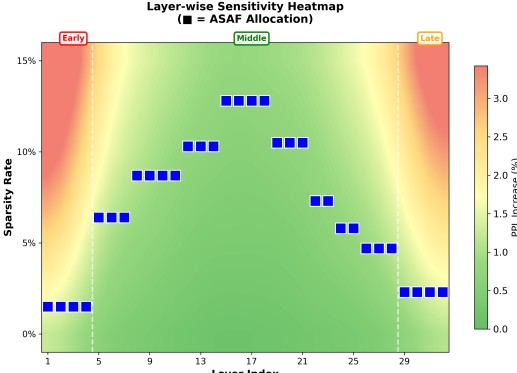

Figure 13: ASAF's learned sparsity allocation showing consecutive layer grouping with adaptive sparsity rates.

### A.19.2 Layer Sensitivity Analysis

Figure 14 presents the empirical sensitivity validation. We measure per-layer sensitivity by applying pruning to individual layers while keeping others unchanged. The results reveal smooth depth-wise transitions: early/late layers exhibit higher sensitivity (2 to 3% perplexity increase), while mid-depth layers tolerate more aggressive pruning (0.5 to 1% increase). This sensitivity pattern directly explains ASAF's learned allocation in Figure 13, demonstrating that the framework automatically discovers optimal allocations correlated with empirical layer sensitivity.

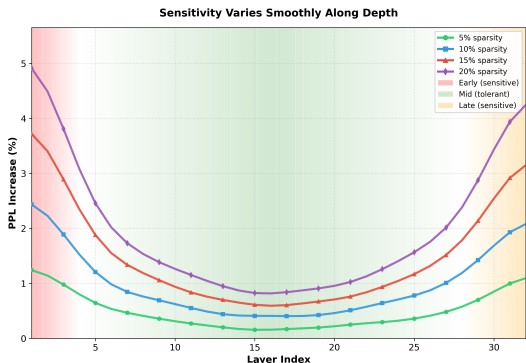

Figure 14: Per-layer sensitivity showing U-shaped pattern that correlates with ASAF's allocation.

## Appendix A.20: Limitations and Future work

**Limitations.** While our empirical study covers multiple Llama-2/3 scales and widely used language benchmarks, it does not exhaust the space of tasks, architectures, or deployment settings. In addition, our dynamic-programming formulation relies on a specific tabulation-based sensitivity proxy and fixed sparsity discretization, which may not capture all nuances of model behavior under compression.

**Future work.** Future work includes extending ASAF to heterogeneous objectives such as latency, energy, or monetary cost, and to a broader family of encoder-only and multimodal LLMs. An-

other direction is combining our layer-wise allocation with structured sparsity or routing schemes to better align compression decisions with downstream scheduling, batching, and real-world serving constraints.