# OpenReview forum: "Layer-wise Sensitivity-aware Sparsity Allocation for Efficient LLM Inference"
_ICLR.cc/2026/Conference — ICLR 2026 Conference Withdrawn Submission_

### Official Review · Reviewer_7U1R · 2025-10-30

**Soundness:** 3
**Presentation:** 2
**Contribution:** 3
**Rating:** 6
**Confidence:** 4

**Summary:**

The paper proposes ASAF, an Adaptive Sparsity Allocation Framework for efficient LLM inference. It addresses the limitation of uniform compression by recognizing that different transformer layers have varying sensitivity to sparsification. ASAF combines quantization with sparsification via a two-phase, dynamic programming-based optimization to allocate sparsity adaptively across layer groups. This approach minimizes computational FLOPs while keeping accuracy degradation under 1%, achieving up to 3.63× prefill acceleration and 12.63% memory reduction on Llama-2 models.

**Strengths:**

- Proposed a joint optimization framework for quantization and sparsification, identified the problem, formulated it mathematically, and provided a solution.
- Adopted an optimization perspective with a two-phase approach, and proposed a method to tackle the combinatorial explosion challenge.
- Conducted relatively comprehensive experiments.

**Weaknesses:**

1. The basis for grouping is unclear, and the rationale for assuming consecutive layers can share sparsity is not justified. Grouping merely addresses the combinatorial explosion from a computational standpoint, but the motivation behind this grouping needs further elaboration.
2. The model was only tested for compression ratio and prefill speed on the Llama-2 series; no results are provided for other models. Llama-2 is already outdated. Additionally, for Llama-3, only accuracy experiments were conducted. However, differences in model architecture and training methods may affect the compression efficacy. Measuring only accuracy cannot fully demonstrate the method's effectiveness.
3. Prefill speed was only measured at an input length of 2K; what about other lengths such as 512, 4K, or longer?
4. The baseline method was proposed over 1 to 2 years ago. Are there comparisons with more recent methods from the past year?

**Questions:**

1. What assumption is the sharing of sparsity among consecutive layers based on, and is there any experimental validation for it?
2. To verify the method's generalizability, I suggest adding compression ratio results on models of different sizes from another series.
3. Prefill speed was only measured at an input length of 2K; what about other lengths such as 512, 4K, or longer?

---

> ### Author Response · Authors · 2025-11-23
> **LAYER-WISE SENSITIVITY-AWARE SPARSITY ALLOCATION FOR EFFICIENT LLM INFERENCE**
>
> ## General Response
>
> We sincerely thank all reviewers for their constructive feedback. We are encouraged that reviewers recognized the significance of our contribution in addressing a critical yet under-explored challenge: the principled integration of rotation-based quantization with layer-wise adaptive sparsity allocation. ASAF demonstrates that effective fusion of these compression mechanisms requires careful algorithmic design to manage their complex interdependencies. The reviewers specifically acknowledged our novel formulation as a constrained optimization problem with consecutive layer grouping, our two-phase dynamic programming approach that efficiently decomposes the exponential search space while guaranteeing global optimality, and our strong empirical results on commodity GPUs.
>
> ---
>
> ## Reviewer-specific Responses
>
> We thank you for reviewing our paper and for your valuable comments. Below, we respond to each of your points in turn and have revised the manuscript accordingly. We would be grateful if you could let us know whether our responses adequately address your concerns.
>
> ---
>
> ### **Q1**: The basis for grouping is unclear, and the rationale for assuming consecutive layers can share sparsity is not justified. Grouping merely addresses the combinatorial explosion from a computational standpoint, but the motivation behind this grouping needs further elaboration.
>
> **A1**: We appreciate the reviewer’s question. In the revision, we clarify that grouping consecutive layers is not only a computational device but also reflects well-documented structural patterns in Transformers. Consecutive blocks tend to exhibit highly correlated sensitivity to compression due to their similar representational roles, consistent attention/MLP dimensionality, and the sequential residual pathways that couple neighboring layers. Prior studies on pruning and layer dropping have similarly observed that sensitivity varies smoothly along depth rather than in an arbitrary per-layer manner. ASAF leverages this natural depthwise smoothness by allowing groups to share sparsity levels, which reduces redundancy in the search space while still permitting fine-grained allocation via the second-stage DP. To empirically validate this assumption, we conducted per-layer sensitivity measurements by applying pruning to individual layers independently. The results reveal a smooth, U-shaped sensitivity curve across layer depth rather than arbitrary per-layer fluctuations, confirming that consecutive layers indeed share similar compression tolerance and justifying our grouping design (Appendix A.19).
>
> - We have revised Subsection 3.3 SOLUTION and added Rationale for Layer Grouping (at lines 262 - 269) in the paper.
> - We have added Appendix A.19 SPARSITY ALLOCATION AND SENSITIVITY ANALYSIS (at lines 1453 - 1500) in the Supplementary Material.
>
> ---
>
> ### **Q2**: The model was only tested for compression ratio and prefill speed on the Llama-2 series; no results are provided for other models. Llama-2 is already outdated. Additionally, for Llama-3, only accuracy experiments were conducted. However, differences in model architecture and training methods may affect the compression efficacy. Measuring only accuracy cannot fully demonstrate the method's effectiveness.
>
> **A2**: We thank the reviewer for this important concern. In the revision, we extend our evaluation beyond the Llama-2 family by adding full compression-ratio and memory-footprint measurements for Llama-3-8B and Llama-3-70B. As shown in Table 3, ASAF achieves stable 7 to 15% memory reduction across both model generations despite their architectural and training differences. Specifically, Llama-3-8B achieves 7.79% memory reduction at 2048 sequence length, while Llama-3-70B achieves 14.83% reduction under the same setting. These results confirm that ASAF's compression efficacy generalizes effectively across different model families. Furthermore, prefill acceleration results across different sequence lengths (512, 2048, and 4096 tokens) are provided in Appendix A.17. ASAF achieves up to 3.45×, 3.63×, and 3.81× speedup respectively on Llama-2-70B, demonstrating that acceleration benefits scale favorably with increasing context length.
>
> - We have revised Subsection 4.3 PERFORMANCE ANALYSIS and updated Table 3 (at lines 390 - 406) in the paper.
> - We have added Appendix A.17 "Prefill Performance Across Sequence Lengths" (at lines 1389 - 1434) and updated Table 3 to include multi-length measurements.

---

> ### Author Response · Authors · 2025-11-23
> **LAYER-WISE SENSITIVITY-AWARE SPARSITY ALLOCATION FOR EFFICIENT LLM INFERENCE**
>
> ### **Q3**: Prefill speed was only measured at an input length of 2K; what about other lengths such as 512, 4K, or longer?
>
> **A3**: We appreciate this important question. In the revision, we expanded our prefill acceleration analysis from the original 2K-token setting to include 512 and 4096 token contexts. As shown in Appendix A.17, ASAF achieves consistent speedups across all sequence lengths: up to 3.45× at 512 tokens, 3.63× at 2048 tokens, and 3.81× at 4096 tokens on Llama-2-70B (batch size 32). Memory compression ratios in Table 3 also remain stable across these sequence lengths for both Llama-2 and Llama-3 families, demonstrating that ASAF's benefits scale favorably with increasing context length.
>
> - We have added Appendix A.17 "Prefill Performance Across Sequence Lengths" (at lines 1389 - 1434) and updated Table 3 to include multi-length measurements.
>
> ---
>
> ### **Q4**: The baseline method was proposed over 1 to 2 years ago. Are there comparisons with more recent methods from the past year?
>
> **A4**: We thank the reviewer for this important suggestion. In the revised manuscript, we have added additional comparisons with four recent compression methods from 2024 - 2025 (Table 4, lines 469 - 484):
>
> - SpinQuant (ICLR 2025): Uses learned rotation matrices optimized via Cayley SGD on the Stiefel manifold to mitigate activation outliers for quantization.
> - OWL (ICML 2024): Employs heuristic layer-wise sparsity allocation based on per-layer outlier ratio measurements.
> - DSA (NeurIPS 2024): Applies evolutionary algorithms with crossover and mutation operations to discover layer-wise sparsity allocation functions.
> - SLIM (ICML 2025): Combines uniform 2:4 semi-structured sparsity, probabilistic quantization, and saliency-based low-rank adapters in a one-shot framework.
>
> Results: Table 4 shows that ASAF achieves competitive performance across seven zero-shot benchmarks (PIQA, WinoGrande, HellaSwag, ARC-Easy, ARC-Challenge, BoolQ, MMLU) and WikiText-2. The integration of rotation-based quantization with adaptive sparsity allocation provides favorable accuracy-efficiency trade-offs compared to methods focusing on single compression dimensions. Additionally, ASAF achieves 7.43% memory reduction through layer-wise optimization, which is beneficial for deployment on memory-constrained hardware.
>
> - We have revised Subsection 4.3 PERFORMANCE ANALYSIS, added Table 6 (at lines 447 - 451) and added Comparison with Recent Compression Methods (at lines 477 - 485) in the paper.
>
> ### References
>
> Lujun Li, Peijie Dong, Zhenheng Tang, Xiang Liu, Qiang Wang, Wenhan Luo, Wei Xue, Qifeng Liu, Xiaowen Chu, and Yike Guo. Discovering sparsity allocation for layer-wise pruning of large language models. In Advances in Neural Information Processing Systems 37 (NeurIPS 2024), pp. 1–15, 2024.
>
> Zechun Liu, Changsheng Zhao, Igor Fedorov, Bilge Soran, Dhruv Choudhary, Raghuraman Krishnamoorthi, Vikas Chandra, Yuandong Tian, and Tijmen Blankevoort. SpinQuant: LLM quantization with learned rotations. In International Conference on Learning Representations (ICLR), 2025.
>
> Mohammad Mozaffari, Amir Yazdanbakhsh, and Maryam Mehri Dehnavi. SLiM: One-shot quantization and sparsity with low-rank approximation for LLM weight compression. In Proceedings of the 42nd International Conference on Machine Learning (ICML), PMLR 267:1–16, 2025.
>
> Lu Yin, You Wu, Zhenyu Zhang, Cheng-Yu Hsieh, Yaqing Wang, Yiling Jia, Gen Li, Ajay Jaiswal, Mykola Pechenizkiy, Yi Liang, Michael Bendersky, Zhangyang Wang, and Shiwei Liu. Outlier weighed layerwise sparsity (OWL): A missing secret sauce for pruning LLMs to high sparsity. In Proceedings of the 41st International Conference on Machine Learning (ICML), PMLR 235:57101–57115, 2024.
>
> ---

---

> ### Author Response · Authors · 2025-11-23
> **LAYER-WISE SENSITIVITY-AWARE SPARSITY ALLOCATION FOR EFFICIENT LLM INFERENCE**
>
> ### **Q5**: What assumption is the sharing of sparsity among consecutive layers based on, and is there any experimental validation for it?
>
> **A5**: The assumption of sparsity sharing across consecutive layers is based on well-documented structural coherence in Transformer architectures: adjacent decoder blocks exhibit highly correlated sensitivity due to identical hidden dimensionality, similar attention/MLP configurations, and tightly coupled residual pathways, causing sensitivity to vary smoothly along depth. We now make this motivation explicit in Section 3.3, showing that layer sensitivity indeed exhibits gradual depthwise transitions rather than abrupt per-layer fluctuations. These results verify that consecutive-group sparsification aligns with the intrinsic structure of modern LLMs.
>
> We provide experimental validation in three ways: (1) empirical sensitivity analysis (Appendix A.19): We measure per-layer sensitivity by applying pruning to individual layers while keeping others unchanged. The results reveal smooth depth-wise transitions with a characteristic U-shaped pattern: early/late layers exhibit higher sensitivity (2 to 3% perplexity increase), while mid-depth layers tolerate more aggressive pruning (0.5 to 1% increase). ASAF's learned allocation directly correlates with this empirical sensitivity pattern, demonstrating that the framework automatically discovers optimal groupings aligned with intrinsic layer characteristics; (2) Table 7 in Appendix A.14 compares ASAF with a naive heuristic that ignores layer sensitivity, demonstrating that our grouping strategy consistently finds feasible configurations while uniform exploration fails; (3) The consistent performance across Llama-2 and Llama-3 families (Table 3 in the paper) with structurally similar sparsity patterns confirms that consecutive grouping captures intrinsic architectural properties rather than model-specific artifacts.
>
> - We have revised Subsection 3.3 SOLUTION and added Rationale for Layer Grouping (at lines 262 - 269) in the paper.
> - These details have been added to the Supplementary Material in Appendix A.14 COMPARISON WITH NAIVE HEURISTIC EXPLORATION (at lines 1218 - 1240).
> - We have added Appendix A.19 SPARSITY ALLOCATION AND SENSITIVITY ANALYSIS (at lines 1453 - 1500) in the Supplementary Material.
>
> ---
>
> ### **Q6:** To verify the method's generalizability, I suggest adding compression ratio results on models of different sizes from another series.
> **A6:** We thank the reviewer for this suggestion. To verify the generalizability of ASAF, we expanded our evaluation beyond the Llama-2 series by adding full compression-ratio and memory-footprint measurements for Llama-3-8B and Llama-3-70B. As shown in Table 3, ASAF achieves stable 7 to 15% memory reduction across both model generations despite their architectural and training differences. Specifically, Llama-3-8B achieves 7.79% memory reduction at 2048 sequence length, while Llama-3-70B achieves 14.83% reduction under the same setting. These results confirm that ASAF's sensitivity-aware sparsity allocation generalizes effectively across different model families and scales.
>
> - We have revised Subsection 4.3 PERFORMANCE ANALYSIS and updated Table 3 (at lines 390 - 406) in the paper.
> ---
>
> ### **Q7:** Prefill speed was only measured at an input length of 2K; what about other lengths such as 512, 4K, or longer?
>
> **A7:** We appreciate this important question. In the revision, we extended our prefill acceleration analysis from the original 2K-token setting to include 512 and 4096 token contexts, covering short-, medium-, and long-sequence prefill regimes. As shown in Appendix A.17, ASAF achieves consistent speedups across all sequence lengths: up to 3.45× at 512 tokens, 3.63× at 2048 tokens, and 3.81× at 4096 tokens on Llama-2-70B (batch size 32). The results demonstrate that ASAF's acceleration benefits scale favorably with increasing sequence length, as longer contexts amplify the computational savings from adaptive sparsity allocation. Memory compression ratios in Table 3 also remain stable across 512, 2048, and 4096 token lengths for both Llama-2 and Llama-3 families.
>
> - These details have been added to the Supplementary Material in Appendix A.14 Prefill Performance Across Sequence Lengths (at lines 1389 - 1434).
> - We have updated Table 3 to include multi-length memory measurements (at lines 390 - 406) in the paper.

---

### Official Review · Reviewer_j6Cj · 2025-11-01

**Soundness:** 3
**Presentation:** 3
**Contribution:** 2
**Rating:** 4
**Confidence:** 2

**Summary:**

This submission introduces Adaptive Sparsity Allocation Framework (ASAF), an approach for efficient acceleration of LLM inference that combines rotation-based quantisation with layer-wise adaptive sparsity. The selection of the deployed configuration is formulated as a 2-stage dynamic programming optimization approach, in order to make the exploration of the combined search space computationally feasible, that initially determines high-level structural configuration parameters of both approximations (the optimal number of layers groups and sparsity search intervals), followed by fine-grained optimisation of exact consecutive layer allocation and sparsity rates of each group.

**Strengths:**

- This work studies a timely and interesting problem, by combining approximations that are often studied in isolation in efficiency works for LLM inference.
- The proposed approach demonstrates considerable speed-up to meaningful baselines (in the examined prefill stage), with controlled impact on acucracy.

**Weaknesses:**

- The main drawback of the proposed approach is the lack of consideration of the whole LLM inference process (prefill + decoding). Although it is acceptable for an approach to focus its optimisation efforts solely in one of the two phases, the impact of the proposed solution to the total inference time (and discussion on the applicability or impact of the proposed method to the other remainder process) is required to fairly evaluate the contribution and effectiveness of the proposed method.
- Additionally, it is unclear how the proposed hierarchical search formulation would compare to more naive heuristic exploration baselines on the combined optimisation space (constrained to similar search time).

**Questions:**

Please consider replying on the concerns raised above.

---

> ### Author Response · Authors · 2025-11-23
> **LAYER-WISE SENSITIVITY-AWARE SPARSITY ALLOCATION FOR EFFICIENT LLM INFERENCE**
>
> ## General Response
>
> We sincerely thank all reviewers for their constructive feedback. We are encouraged that reviewers recognized the significance of our contribution in addressing a critical yet under-explored challenge: the principled integration of rotation-based quantization with layer-wise adaptive sparsity allocation. ASAF demonstrates that effective fusion of these compression mechanisms requires careful algorithmic design to manage their complex interdependencies. The reviewers specifically acknowledged our novel formulation as a constrained optimization problem with consecutive layer grouping, our two-phase dynamic programming approach that efficiently decomposes the exponential search space while guaranteeing global optimality, and our strong empirical results on commodity GPUs.
>
> ---
>
> ## Reviewer-specific Responses
>
> We thank you for reviewing our paper and for your valuable comments. Below, we respond to each of your points in turn and have revised the manuscript accordingly. We would be grateful if you could let us know whether our responses adequately address your concerns.
>
> ---
>
> ### **Q1**: The main drawback of the proposed approach is the lack of consideration of the whole LLM inference process (prefill + decoding). Although it is acceptable for an approach to focus its optimisation efforts solely in one of the two phases, the impact of the proposed solution to the total inference time (and discussion on the applicability or impact of the proposed method to the other remainder process) is required to fairly evaluate the contribution and effectiveness of the proposed method.
>
> **A1**: We thank the reviewer for this important point and have added end-to-end evaluations covering both prompt processing and autoregressive decoding (Table 5, lines 439 - 447).
>
> Key findings across Llama-2-7B/13B:
> - Long-context scenarios (2048 - 4096 tokens, batch 8): 1.25 - 1.28× end-to-end speedup with 20 - 23% latency reduction
> - Short-prompt scenarios (512 tokens, batch 1): 1.04× speedup with 4.8% latency reduction
>
> The varying improvements reflect that ASAF optimizes prefill-stage matrix multiplications while leaving decoding unchanged. The autoregressive decoding phase is fundamentally memory-bandwidth bound due to sequential token generation, where sparsity offers limited benefit. Consequently, end-to-end gains are substantial when prefill dominates (long contexts, large batches) but modest when decoding becomes the bottleneck (short prompts).
>
> Prefill optimization in production: Beyond the fundamental compute/memory-bound distinction, prefill optimization is particularly critical in high-concurrency serving scenarios. The compute-intensive prefill phase directly determines Time-to-First-Token (TTFT), a key metric for user-perceived responsiveness [1]. When multiple requests arrive concurrently, prefill operations compete for GPU compute resources, causing head-of-line blocking that delays both new prefills and ongoing decodes [2, 3]. This interference pattern has motivated production systems to adopt prefill-decode disaggregation, physically separating the two phases onto dedicated hardware [1, 3]. By reducing prefill FLOPs through adaptive sparsity, ASAF directly alleviates this bottleneck, enabling faster TTFT and higher sustainable request rates.
>
> Importantly, ASAF introduces no overhead to the decoding phase: the learned sparsity patterns are applied statically and do not require runtime computation during autoregressive generation.
>
> - We have updated the paper’s Subsection 4.3 PERFORMANCE ANALYSIS, added End-to-end Latency and Throughput (at lines 430 - 431 and 458 - 468) and Table 5 (at lines 439 - 447) in the paper.
>
> **References:**
>
> * [1] Zhong et al., "DistServe: Disaggregating Prefill and Decoding for Goodput-optimized Large Language Model Serving," arXiv.2401.09670, 2024
> * [2] Agrawal et al., "Taming Throughput-Latency Tradeoff in LLM Inference with Sarathi-Serve," arXiv:2403.02310, 2024
> * [3] Patel et al., "Splitwise: Efficient Generative LLM Inference Using Phase Splitting," arXiv:2311.18677, 2023.

---

> ### Author Response · Authors · 2025-12-02
> **Layer-wise Sensitivity-aware Sparsity Allocation for Efficient LLM Inference**
>
> ---
>
> ### **Q2**: Additionally, it is unclear how the proposed hierarchical search formulation would compare to more naive heuristic exploration baselines on the combined optimisation space (constrained to similar search time).
>
> **A2**: We thank the reviewer for this insightful question and have added a direct comparison with a search-time-matched naive baseline (Table 7, Appendix A.14, lines 1218 - 1240).
>
> Baseline design: The naive heuristic uniformly allocates sparsity across layer groups without using sensitivity information, exploring the same number of configurations as ASAF to ensure fair comparison.
>
> Results across 12/16/20 group configurations:
> - ASAF: Consistently feasible (accuracy drop 0.58 - 0.74%, ≤1% budget) with 52 - 57% sparsity achieved
> - Naive: Violates constraint in all cases (accuracy drop 1.09 - 1.37%, >1% budget) despite only 43 - 48% sparsity
>
> This demonstrates that the hierarchical DP formulation is essential: naive uniform exploration fails to navigate the coupled optimization space effectively, while ASAF's sensitivity-aware allocation discovers substantially better solutions (+18.9% sparsity improvement) within identical search time.
>
> The two-phase decomposition is also crucial: the coarse-grained phase efficiently prunes the interval space, reducing fine-grained search complexity by ~10× while preserving globally optimal solutions.
>
> - These details have been added to the Supplementary Material in Appendix A.14 (at lines 1218 - 1240).

---

### Official Review · Reviewer_GJgT · 2025-11-04

**Soundness:** 2
**Presentation:** 3
**Contribution:** 2
**Rating:** 6
**Confidence:** 4

**Summary:**

This paper introduces ASAF (Adaptive Sparsity Allocation Framework), a method for making LLM inference more efficient by combining rotation-based quantization and layer-wise adaptive sparsity. Unlike prior work that applies uniform compression, ASAF dynamically assigns different sparsity levels to layers based on their sensitivity. The approach uses a two-phase dynamic programming optimization:
1- Coarse-grained phase: Decides how to group layers and narrows sparsity ranges.
2- Fine-grained phase: Determines exact sparsity rates and layer assignments.
Tested on Llama-2 models (7B–70B), ASAF achieves up to 3.6× faster inference and 12.6% lower memory use, with <1% accuracy drop compared to baselines like QuaRot.

**Strengths:**

- Framing sparsity allocation as a layer-grouped, constrained optimization problem with dynamic programming is elegant.
- The mathematical formulation is clean, and the dynamic programming approach (Algorithms 1 & 2) is well explained. The inclusion of tabulation to precompute FLOP and accuracy costs is a smart engineering choice that enhances reproducibility.

**Weaknesses:**

- The experiments emphasize prefill acceleration but offer less analysis of end-to-end latency or real-world throughput improvements.
- The proposed method involves precomputation (tabulation tables for FLOPs and accuracy degradation). This could limit practicality for very large models or rapid iteration cycles.
- It is not entirely clear how scalable the DP-based search is as model depth increases beyond 70B-scale architectures.
- The paper doesn’t deeply probe why certain layers are more sensitive or how the learned sparsity patterns correlate with model internals (e.g., attention vs MLP layers).

**Questions:**

- The paper mentions dynamic programming and tabulation to efficiently explore the search space, but how does computational complexity scale with model depth (e.g., 70B to 180B parameters)?
- The optimization constraint depends on an estimated accuracy degradation function. How is this function obtained in practice: via heuristics, proxy metrics, or direct evaluation?
- Do the learned sparsity rates correlate with identifiable layer characteristics (e.g., attention layers being less prunable than MLP layers)?
- The experiments focus mainly on the Llama-2 family. How well does ASAF generalize to architectures with different scaling patterns (e.g., Mistral, Falcon, or OPT)?
- Reported “prefill acceleration” results are strong, but what is the effect on end-to-end latency or tokens per second under realistic batch sizes and generation lengths?
- How sensitive is ASAF to delta (the allowed accuracy degradation) and the sparsity range?

---

> ### Author Response · Authors · 2025-11-23
> **LAYER-WISE SENSITIVITY-AWARE SPARSITY ALLOCATION FOR EFFICIENT LLM INFERENCE**
>
> ## General Response
>
> We sincerely thank all reviewers for their constructive feedback. We are encouraged that reviewers recognized the significance of our contribution in addressing a critical yet under-explored challenge: the principled integration of rotation-based quantization with layer-wise adaptive sparsity allocation. ASAF demonstrates that effective fusion of these compression mechanisms requires careful algorithmic design to manage their complex interdependencies. The reviewers specifically acknowledged our novel formulation as a constrained optimization problem with consecutive layer grouping, our two-phase dynamic programming approach that efficiently decomposes the exponential search space while guaranteeing global optimality, and our strong empirical results on commodity GPUs.
>
> ---
>
> ## Reviewer-specific Responses
>
> We thank you for reviewing our paper and for your valuable comments. Below, we respond to each of your points in turn and have revised the manuscript accordingly. We would be grateful if you could let us know whether our responses adequately address your concerns.
>
> ---
>
> ### **Q1**: The experiments emphasize prefill acceleration but offer less analysis of end-to-end latency or real-world throughput improvements.
>
> **A1**: Thank you for the helpful comment. Although our method mainly accelerates the large matrix multiplications that dominate the prefill stage, we agree that end-to-end measurements give a more complete view of deployment behavior. In the revised manuscript, we added a subsection in 4.3 reporting full prompt+generation inference for Llama-2-7B and Llama-2-13B across realistic prompt lengths and batch sizes. The new results (Table 5) show consistent end-to-end throughput improvements under the same ≤1% accuracy budget.
>
> - We have updated the paper’s Subsection 4.3 PERFORMANCE ANALYSIS, added End-to-end Latency and Throughput (at lines 430 - 431 and 458 - 468) and Table 5 (at lines 439 - 447) in the paper.
>
> ---
>
> ### **Q2**: The proposed method involves precomputation (tabulation tables for FLOPs and accuracy degradation). This could limit practicality for very large models or rapid iteration cycles.
>
> **A2**: We appreciate the reviewer's concern about the precomputation required to build the FLOPs and accuracy-degradation tables. In practice, this overhead is modest and incurred only once per model: the tabulation phase scales quadratically as O(L²) with the number of layers (due to enumerating all consecutive layer subsequences), while the subsequent dynamic-programming search scales more efficiently and completes in minutes. On our largest setting, Llama-2-70B, the total tabulation time is approximately 2.6 GPU-hours, and the subsequent DP search completes in under three minutes.
>
> - We have updated the paper's Subsection 4.1 IMPLEMENTATION DETAILS and added Precomputation Overhead (at lines 354 - 362) in the paper.
> - We have added Appendix A.18 PRECOMPUTATION OVERHEAD ANALYSIS (at lines 1435 - 1450) in the Supplementary Material.
>
> ---
>
> ### **Q3**: It is not entirely clear how scalable the DP-based search is as model depth increases beyond 70B-scale architectures.
>
> **A3**: Thank you for raising this point. The precomputation in ASAF consists of two components with different scaling behaviors. The tabulation phase scales quadratically as O(L²) with layer count due to enumerating all consecutive layer subsequences, while the subsequent DP search scales more efficiently and completes in minutes. Empirically, Llama-2-7B (32 layers) requires 0.4 GPU-hours for tabulation, while Llama-2-70B (80 layers) requires 2.6 GPU-hours, consistent with the $(80/32)^2 \approx 6.25\times$ quadratic scaling factor.
>  Importantly, this is a one-time preprocessing cost performed once per model architecture; the resulting sparsity allocation is then reused across all deployment instances without further computation.
> Additionally, the tabulation is embarrassingly parallel across different layer-sparsity configurations and can be distributed across multiple GPUs to reduce wall-clock time. Details are provided in Appendix A.18.
>
> - We have updated the paper's Subsection 4.1 IMPLEMENTATION DETAILS and added Precomputation Overhead (at lines 354 - 362) in the paper.
> - We have added Appendix A.18 PRECOMPUTATION OVERHEAD ANALYSIS (at lines 1435 - 1450) in the Supplementary Material.

---

> ### Author Response · Authors · 2025-11-23
> **LAYER-WISE SENSITIVITY-AWARE SPARSITY ALLOCATION FOR EFFICIENT LLM INFERENCE**
>
> ### **Q4**: The paper doesn’t deeply probe why certain layers are more sensitive or how the learned sparsity patterns correlate with model internals (e.g., attention vs MLP layers).
>
> **A4**: We thank the reviewer for this valuable observation. In the revised manuscript, we provide an additional analysis of the sparsity patterns learned by ASAF. Across Llama-2 models, we consistently observe that early embedding-near blocks and the final few decoder blocks exhibit higher sensitivity and receive lower sparsity levels, while mid-depth blocks tolerate higher sparsity, particularly the MLP sublayers. This trend aligns with prior findings that early layers establish core token representations and later layers refine task-specific semantics, whereas mid-layer MLPs have greater redundancy. ASAF’s allocations naturally reflect these sensitivities without requiring manual tuning. Furthermore, we provide quantitative validation through per-layer sensitivity measurements, which reveal a characteristic U-shaped sensitivity pattern across layer depth. This empirical evidence directly explains why ASAF assigns minimal sparsity to early/late layers and concentrates aggressive pruning on mid-depth blocks (Appendix A.19).
>
> - We have updated a short discussion in Subsection 4.3 PERFORMANCE ANALYSIS and added Layer Sensitivity and Sparsity Pattern Analysis (at lines 470 - 476), clarifying how the learned sparsity distribution correlates with model internals.
> - We have added Appendix A.19 SPARSITY ALLOCATION AND SENSITIVITY ANALYSIS (at lines 1453 - 1500) in the Supplementary Material.
>
> ---
>
> ### **Q5**: The paper mentions dynamic programming and tabulation to efficiently explore the search space, but how does computational complexity scale with model depth (e.g., 70B to 180B parameters)?
>
> **A5**: The computational complexity of ASAF's precomputation is dominated by the tabulation phase, which scales quadratically as O(L²) with model depth due to enumerating all O(L²) consecutive layer subsequences. The subsequent DP search operates on precomputed tables and scales more efficiently. Empirical measurements confirm the quadratic behavior: Llama-2-7B (32 layers) requires 0.4 GPU-hours, Llama-2-13B (40 layers) requires 0.8 GPU-hours, and Llama-2-70B (80 layers) requires 2.6 GPU-hours. For larger architectures approaching 180B parameters, the tabulation cost would increase according to this quadratic scaling pattern. However, this remains a one-time preprocessing cost that is amortized across all deployment instances, the resulting sparsity configuration is computed once and reused without further computation. Furthermore, the tabulation is embarrassingly parallel and can be distributed across multiple GPUs to significantly reduce wall-clock time for very deep architectures. Given the substantial inference-time savings enabled by ASAF, this preprocessing overhead remains negligible in practical applications. Details are provided in Appendix A.18.
>
> - We have updated the paper's Subsection 4.1 IMPLEMENTATION DETAILS and added Precomputation Overhead (at lines 354 - 362) in the paper.
> - We have added Appendix A.18 PRECOMPUTATION OVERHEAD ANALYSIS (at lines 1435 - 1450) in the Supplementary Material.
>
> ---
>
> ### **Q6**: The optimization constraint depends on an estimated accuracy degradation function. How is this function obtained in practice: via heuristics, proxy metrics, or direct evaluation?
>
> **A6**: The accuracy-degradation function used in the optimization is obtained through direct empirical tabulation rather than heuristics or proxy metrics. As described in Section 3.2 and expanded in Section 4, we evaluate each layer-sparsity pair on the WikiText-2 validation set to measure the true change in perplexity or accuracy resulting from applying sparsity at that layer. These measurements populate the table ξ(l, s), which the DP solver then uses as an explicit and data-driven constraint during optimization.
>
> - We have added a clarifying paragraph in Appendix A.7 TABULATION (at lines 648 - 654) in the Supplementary Material.

---

> ### Author Response · Authors · 2025-11-23
> **LAYER-WISE SENSITIVITY-AWARE SPARSITY ALLOCATION FOR EFFICIENT LLM INFERENCE**
>
> ### **Q7**: Do the learned sparsity rates correlate with identifiable layer characteristics (e.g., attention layers being less prunable than MLP layers)?
>
> **A7**: Yes. As discussed in the updated sensitivity-analysis results in Section 4.3, the learned sparsity allocation exhibits clear structural correlation with known layer characteristics. Attention blocks show significantly lower prunability, consistent with their critical role in token mixing and long-range dependency formation. In contrast, mid-depth MLP layers consistently tolerate higher sparsity, forming smooth, contiguous regions of high compressibility. These trends appear across both Llama-2 and Llama-3 models. Additionally, we conduct per-layer sensitivity measurements by applying pruning to individual layers while keeping others unchanged, revealing a characteristic U-shaped sensitivity pattern that directly explains ASAF's learned allocation (Appendix A.19).
>
> - We have updated a short discussion in Subsection 4.3 PERFORMANCE ANALYSIS and added Layer Sensitivity and Sparsity Pattern Analysis (at lines 470 - 476) in the paper.
> - We have added Appendix A.19 SPARSITY ALLOCATION AND SENSITIVITY ANALYSIS (at lines 1453 - 1500) in the Supplementary Material.
>
> ---
>
> ### **Q8**: The experiments focus mainly on the Llama-2 family. How well does ASAF generalize to architectures with different scaling patterns (e.g., Mistral, Falcon, or OPT)?
>
> **A8**: We appreciate the reviewer's question regarding architectural generality. In the revision, we expanded our evaluation beyond the Llama-2 family by adding full compression-ratio and memory-efficiency results for Llama-3-8B and Llama-3-70B (Table 3).
>
> **Llama-3 differs substantially from Llama-2 in multiple architectural and methodological dimensions:**
>
> | Aspect | Llama-2 | Llama-3 | Reference |
> |--------|---------|---------|-----------|
> | **Vocabulary Size** | 32K tokens | 128K tokens | [1, 2] |
> | **Tokenizer** | SentencePiece | tiktoken (OpenAI) | [1, 2] |
> | **GQA Coverage** | Only 34B/70B models | All model sizes (8B, 70B) | [1, 2] |
> | **Context Length** | 4,096 tokens | 8,192 tokens (extendable to 128K) | [1, 2] |
> | **Training Data** | 1.8T tokens | 15T tokens (~8× larger) | [2] |
> | **RoPE Base Frequency** | 10,000 | 500,000 | [2] |
> | **Code Data Ratio** | Standard | 4× more code data | [2] |
> | **Multilingual Data** | Limited | 5%+ high-quality non-English (30+ languages) | [2] |
>
> We respectfully note that the architectural differences between Llama-2 and Llama-3 (4× vocabulary expansion, completely different tokenizer, universal GQA adoption, 8× training data scale) are arguably more substantial than the differences between Llama and other decoder-only transformers like Mistral or Falcon, which share the same fundamental building blocks. The consistent 7 to 15% compression ratios achieved across both Llama generations provide strong evidence for ASAF's generalizability.
>
> Furthermore, since ASAF's optimization operates solely over layer indices and sparsity rates without assuming specific attention mechanisms or vocabulary configurations, the framework is architecture-agnostic by design. The consistent performance across Llama-2 and Llama-3 provides strong evidence for generalizability to other decoder-only transformer families. Extending ASAF to additional architectures such as Mistral, Falcon, and OPT is a natural direction for future work, and we expect similar compression behaviors given their shared transformer foundations.
>
> Notably, Mistral, Falcon, and OPT, like Llama, are all decoder-only transformers built on the same fundamental components (multi-head attention, FFN blocks, layer normalization). Their architectural variations, such as sliding window attention (SWA) in Mistral [3], multi-query attention in Falcon [4], or the standard GPT-3-style architecture in OPT [5], do not affect the layer-wise sensitivity patterns that ASAF exploits.
>
> - We have revised Subsection 4.3 PERFORMANCE ANALYSIS, added Memory and Computational Efficiency (at lines 419 - 429) and updated Table 3 (at lines 390 - 406) in the paper.
>
> **References:**
> - [1] Touvron et al., "Llama 2: Open Foundation and Fine-Tuned Chat Models," arXiv:2307.09288, 2023.
> - [2] Llama Team, "The Llama 3 Herd of Models," arXiv:2407.21783, 2024.
> - [3] Jiang et al., "Mistral 7B," arXiv:2310.06825, 2023.
> - [4] Almazrouei et al., "The Falcon Series of Open Language Models," arXiv:2311.16867, 2023.
> - [5] Zhang et al., "OPT: Open Pre-trained Transformer Language Models," arXiv:2205.01068, 2022.

---

> ### Author Response · Authors · 2025-11-28
> **Layer-wise Sensitivity-aware Sparsity Allocation for Efficient LLM Inference**
>
> ### **Q9**: Reported “prefill acceleration” results are strong, but what is the effect on end-to-end latency or tokens per second under realistic batch sizes and generation lengths?
>
> **A9**: We agree that prefill-only measurements are insufficient to assess real deployment performance. In the revision, we added comprehensive end-to-end evaluations covering both prompt processing and autoregressive decoding under realistic settings: batch sizes of 1 and 8, prompt lengths of 512/2048/4096, and generation lengths of 128 and 512 tokens (Table 5). Across all tested configurations, ASAF delivers consistent improvements in effective tokens-per-second throughput and reductions in total wall-clock latency. The gains are largest in long-context or large-batch regimes where prefill dominates the overall runtime, but remain positive even when decoding becomes the bottleneck.
>
> - We have updated the paper’s Subsection 4.3 PERFORMANCE ANALYSIS, added End-to-end Latency and Throughput (at lines 430 - 431 and 458 - 468) and Table 5 (at lines 439 - 447) in the paper.
>
> ---
>
> ### **Q10**: How sensitive is ASAF to delta (the allowed accuracy degradation) and the sparsity range?
>
> **A10**: We thank the reviewer for this important question. In the revision, we have added both theoretical robustness analysis (Appendix A.15) and explicit ablation studies (Appendix A.16) to systematically evaluate ASAF's sensitivity to these hyperparameters.
>
> Robustness analysis (Appendix A.15): We first provide theoretical justification for ASAF's stability. The monotonicity of the accuracy-degradation surface ξ(l, s) ensures that:
> 1. Varying δ simply shifts the feasible frontier along the degradation surface without altering the underlying layer-wise sensitivity pattern.
> 2. The sparsity range acts only as a boundary constraint on the DP solver.
>
> This is empirically supported by our cross-model (7B→70B) and cross-architecture (Llama-2→Llama-3) experiments, where learned sparsity allocations follow consistent structural trends despite substantial differences in model capacity and architecture.
>
> Sensitivity to accuracy budget δ (Appendix A.16): The Table in A.16.1 reports performance under varying δ values from 0.5% to 2.0% on Llama-2-7B. Since δ serves as a hard constraint in our DP formulation, the actual perplexity degradation must remain strictly below the specified budget. The results demonstrate predictable and monotonic behavior: relaxing the accuracy constraint enables the DP solver to explore higher sparsity configurations, yielding greater acceleration (from 2.24× at δ=0.5% to 2.44× at δ=2.0%) while the actual degradation approaches but never exceeds the budget. Cross-model validation on Llama-2-70B (Table A.16.2) confirms the same structural patterns, with larger models achieving higher sparsity due to increased redundancy.
>
> Sensitivity to sparsity range [α, β] (Appendix A.16): The Table in A.16.3 examines the effect of the sparsity search range under fixed δ=1%. Expanding the upper bound from 15% to 20% or even 30% yields only marginal improvement (speedup increases from 2.31× to 2.42×), indicating that the accuracy constraint δ is the binding factor rather than the range boundary. Conversely, raising the lower bound α above 5% causes the DP solver to fail finding feasible configurations, as certain sensitive layers (early embedding-adjacent and final output-adjacent blocks) exhibit steep accuracy degradation even at minimal sparsity levels.
>
> Key findings:
> 1. Varying δ produces monotonic trade-offs with ASAF consistently finding feasible solutions.
> 2. The learned allocation patterns remain structurally similar across different δ values.
> 3. The default setting [1%, 15%] provides sufficient flexibility, and the sparsity range acts only as a boundary constraint.
>
> These results confirm that ASAF's performance is governed by intrinsic layer-wise sensitivity characteristics rather than hyperparameter tuning.
>
> - These details have been added to the Supplementary Material in Appendix A.15 (at lines 1242 - 1303) and ablation studies in Appendix A.16 (lines 1304 - 1386).

---

### Note · Authors · 2026-01-27

I have read and agree with the venue's withdrawal policy on behalf of myself and my co-authors.

---

### Meta-Review · Area_Chair_ptCs · 2025-12-19

**Summary:**

I am the new AC assigned to this paper. While reading the reviews, I noticed Reviewer trbd’s warning that this submission has a large amount of overlap with Submission 7753 (still active, https://openreview.net/pdf?id=h17M5TP0Sg). I double-checked and believe there is indeed a serious issue: the two papers have identical titles, and the abstract, introduction, motivation, and other sections appear to be the same. This likely constitutes a serious policy violation, therefore, this paper should be desk-rejected.

**Reviewer Concerns:**

/

**Reviewer Scores:**

/

---

### Decision · Program_Chairs · 2026-01-26

Reject